Proceedings of the 7th Symposium on Advances in Approximate Bayesian Inference, 2025 1–24

# Deep Q-Exponential Processes

**Zhi Chang**                                                    ZCHANG7@ASU.EDU

**Chukwudi Paul Obite**                                          COBITE@ASU.EDU

**Shuang Zhou**                                                  SZHOU98@ASU.EDU

**Shiwei Lan** *                                                 SLAN@ASU.EDU

*School of Mathematical & Statistical Sciences, Arizona State University, 901 S Palm Walk, Tempe, AZ 85287, USA*

## Abstract

Motivated by deep neural networks, the deep Gaussian process (DGP) generalizes the standard GP by stacking multiple layers of GPs. Despite the enhanced expressiveness, GP, as an $L_2$ regularization prior, tends to be over-smooth and sub-optimal for inhomogeneous objects, such as images with edges. Recently, Q-exponential process (Q-EP) has been proposed as an $L_q$ relaxation to GP and demonstrated with more desirable regularization properties through a parameter $q > 0$ with $q = 2$ corresponding to GP. Sharing the similar tractability of posterior and predictive distributions with GP, Q-EP can also be stacked to improve its modeling flexibility. In this paper, we generalize Q-EP to deep Q-EP to model inhomogeneous data with improved expressiveness. We introduce shallow Q-EP as a latent variable model and then build a hierarchy of the shallow Q-EP layers. Sparse approximation by inducing points and scalable variational strategy are applied to facilitate the inference. We demonstrate the numerical advantages of the proposed deep Q-EP model by comparing with multiple state-of-the-art deep probabilistic models.

**Keywords:** Deep Probabilistic Models, Inhomogeneous Subjects, Regularization, Latent Representation, Model Expressiveness

## 1. Introduction

Gaussian process (GP Rasmussen and Williams, 2005; J. M. Bernardo and Smith, 1998) has gained enormous successes and been widely used in statistics and machine learning community. With its flexibility in learning functional relationships (Rasmussen and Williams, 2005) and latent representations (Titsias and Lawrence, 2010), and capability in tractable uncertainty quantification, GP has become one of the most popular non-parametric modeling tools. Facilitated by the sparse approximation (Titsias, 2009) and scalable variational inferences (SVGP Hensman et al., 2015; Salimbeni and Deisenroth, 2017), GP has been popularized for a variety of high-dimensional learning tasks. Inspired by the advancement of deep learning (Goodfellow et al., 2016), Damianou and Lawrence (2013) pioneered in generalizing GP with deep structures, hence named deep GP. Ever since then, there has been a large volume of follow-up works including deep convolutional GP (Blomqvist et al., 2020), deep sigma point process (DSPP Jankowiak et al., 2020b), deep image prior (Ulyanov et al., 2020), deep kernel process (Aitchison et al., 2021), deep variational implicit process

---

*Corresponding author.

(Ortega et al., 2023), deep horseshoe GP (Castillo and Randrianarisoa, 2024), and various applications (Dutordoir et al., 2020; Li et al., 2021; Jones et al., 2023).

Despite its flexibility, GP, as an $L_2$ regularization method, tends to produce random candidate functions that are over-smooth and thus sub-optimal for modeling inhomogeneous objects with abrupt changes or sharp contrast. To address this issue, an $L_q$ based stochastic process, $Q$-exponential process (Q-EP Li et al., 2023), has recently been proposed to impose flexible regularization through a parameter $q > 0$, which includes GP as a special case when $q = 2$. Similarly as Lasso inducing sparsity for regression, $q = 1$ is often adopted for Q-EP to impose stronger regularization than GP to properly capture dramatic changes in certain portions of inhomogeneous data, e.g., edges in an image. Different from other $L_1$ based priors such as Laplace random field (Podgórski and Wegener, 2011; Kozubowski et al., 2013) and Besov process (Lassas et al., 2009; Dashti et al., 2012), Q-EP shares with GP the unique tractability of posterior and predictive distributions (Theorem 3.5 of Li et al., 2023), which essentially permits a deep generalization by stacking multiple stochastic mappings (Damianou and Lawrence, 2013).

Motivated by the improved expressiveness of deep GP and the flexible regularization of Q-EP, in this work we generalize Q-EP to *deep Q-EP* to enhance the capability of Q-EP in modeling inhomogeneous data. On one hand, by stacking multiple layers of Q-EP mappings, deep Q-EP becomes more capable of characterizing complex latent representations than the standard Q-EP. On the other hand, inherited from Q-EP, deep Q-EP maintains the control of regularization through the parameter $q > 0$, whose smaller values impose stronger regularization, more amenable than (deep) GP to preserve inhomogenous traits such as edges in an image. First, we introduce the building block, shallow Q-EP model, which can be regarded as a kernelized latent variable model (LVM) (Lawrence, 2003; Titsias and Lawrence, 2010). Such shallow model is also viewed as a stochastic mapping $F$ from input (or latent) variables $X$ to output variables $Y$ defined by a kernel. Then as in Lawrence and Moore (2007); Damianou and Lawrence (2013), we extend such mapping by stacking multiple shallow Q-EP layers to form a hierarchy for the deep Q-EP. Sparse approximation by inducing points (Titsias, 2009) is adopted for the variational inference of deep Q-EP. A theoretic barricade for developing the evidence lower bound (ELBO) in the setting of Q-EP is that the power in the exponent of its density makes involved expectations intractable. We solve this challenge by taking advantage of Jensen's inequality. The inference procedure, as in deep GP, can be efficiently implemented in `GPyTorch` (Gardner et al., 2018).

**Connection to existing works**  Our proposed deep Q-EP is closely related to deep GP (Damianou and Lawrence, 2013) and two other works, deep kernel learning (DKL-GP Wilson et al., 2016) and DSPP (Jankowiak et al., 2020b). Deep Q-EP generalizes deep GP with a parameter $q > 0$ to control the regularization (See Figure 1 for its effect on learning representations) and includes deep GP as a special case for $q = 2$. DKL-GP combines the deep learning architectures (neural networks) with the non-parametric flexibility of kernel methods (GP). DSPP is motivated by parametric GP models (PPGPR Jankowiak et al., 2020a) and applies sigma point approximation or quadrature-like integration to the predictive distribution. The majority of popular deep probabilistic models rely on GP. This is one of the few developed out of a non-Gaussian stochastic process. Our proposed work on deep Q-EP has multi-fold contributions to deep probabilistic models:

1. We propose a novel deep probabilistic model based on Q-EP that generalizes deep GP with flexibility of regularization for handling data inhomogeneity.
2. We develop the variational inference for deep Q-EP and efficiently implement it.
3. We demonstrate numerical advantages of deep Q-EP over its shallow counterpart and the state-of-the-art deep probabilistic models.

The rest of the paper is organized as follows. Section 2 introduces the background of Q-EP. We then develop shallow Q-EP in Section 3 as the building block for deep Q-EP in Section 4. In these two sections, we highlight the importance of posterior tractability in the development and some obstacles in deriving the variational lower bounds. In Section 5 we demonstrate the numerical advantages by comparing with multiple deep probabilistic models in various learning tasks. Finally, we conclude with some discussion on the limitation and potential improvement in Section 6.

## 2. Background: $Q$-exponential Processes

### 2.1. Multivariate $Q$-exponential Distribution

Based on $L_q$ regularization, the univariate *q-exponential distribution* (Dashti et al., 2012) has density $\pi_q(u) \propto \exp\left(-\frac{1}{2}|u|^q\right)$. Li et al. (2023) generalize the univariate $q$-exponential random variable to a multivariate random vector on which a stochastic process can be defined with two requirements by the Kolmogorov' extension theorem (Øksendal, 2003): i) **exchangeability** of the joint distribution, i.e. $p(\mathbf{u}_{1:N}) = p(\mathbf{u}_{\tau(1:N)})$ for any finite permutation $\tau$; and ii) **consistency** of marginalization, i.e. $p(\mathbf{u}_1) = \int p(\mathbf{u}_1, \mathbf{u}_2) d\mathbf{u}_2$.

Suppose a function $u(x)$ is observed at $N$ locations, $x_1, \cdots, x_N \in \mathcal{D} \subset \mathbb{R}^d$. Li et al. (2023) provide a consistent generalization, named *multivariate q-exponential distribution*, for $\mathbf{u} = (u(x_1), \cdots, u(x_N))$ from the family of elliptic contour distributions (Johnson, 1987).

**Definition 1** *A multivariate q-exponential distribution for a random vector $\mathbf{u} \in \mathbb{R}^N$, denoted as* q-ED$_N(\boldsymbol{\mu}, \mathbf{C})$*, has the following density:*

$$p(\mathbf{u}|\boldsymbol{\mu}, \mathbf{C}, q) = \frac{q}{2}(2\pi)^{-\frac{N}{2}}|\mathbf{C}|^{-\frac{1}{2}}r(\mathbf{u})^{(\frac{q}{2}-1)\frac{N}{2}}\exp\left\{-\frac{r^{\frac{q}{2}}}{2}\right\}, \quad r = (\mathbf{u}-\boldsymbol{\mu})^\mathsf{T}\mathbf{C}^{-1}(\mathbf{u}-\boldsymbol{\mu}). \quad (1)$$

**Remark 2** *If taken negative logarithm, the density of* q-ED *in* (1) *yields a quantity dominated by some weighted $L_q$ norm of $\mathbf{u} - \boldsymbol{\mu}$, i.e. $\frac{1}{2}r^{\frac{q}{2}} = \frac{1}{2}\|\mathbf{u}-\boldsymbol{\mu}\|_\mathbf{C}^q$. From the optimization perspective,* q-ED*, when used as a prior, imposes $L_q$ regularization in obtaining the maximum a posteriori (MAP).*

### 2.2. $Q$-exponential Process and Multi-output Q-EP

Li et al. (2023) prove that the multivariate $q$-exponential random vector $\mathbf{u} \sim$ q-ED$_N(\mathbf{0}, \mathbf{C})$ satisfies the conditions of Kolmogorov's extension theorem hence it can be generalized to a stochastic process. For this purpose, we scale it by a factor $N^{\frac{1}{2}-\frac{1}{q}}$ so that its covariance is asymptotically finite (Proposition 3.1 of Li et al., 2023). If $\mathbf{u} \sim$ q-ED$_N(\mathbf{0}, \mathbf{C})$, then we denote $\mathbf{u}^* := N^{\frac{1}{2}-\frac{1}{q}}\mathbf{u} \sim$ q-ED$_N^*(\mathbf{0}, \mathbf{C})$ as a *scaled $q$-exponential random variable*. With a covariance (symmetric and positive-definite) kernel $\mathcal{C} : \mathcal{D} \times \mathcal{D} \to \mathbb{R}$, we define the following *q-exponential process (Q-EP)* based on the scaled $q$-exponential distribution q-ED$_N^*(\mathbf{0}, \mathbf{C})$.

**Definition 3** *A (centered) q-exponential process $u(x)$ with a kernel $\mathcal{C}$, q-$\mathcal{EP}(0,\mathcal{C})$, is a collection of random variables such that any finite set, $\mathbf{u} := (u(x_1), \cdots u(x_N))$, follows a scaled multivariate q-exponential distribution q-ED$^*(\mathbf{0}, \mathbf{C})$, where $\mathbf{C} = [\mathcal{C}(x_i, x_j)]_{N \times N}$. If $\mathcal{C} = \mathcal{I}$, then $u$ is said to be marginally identical but uncorrelated (m.i.u.).*

**Remark 4** *When $q = 2$, q-ED$_N(\boldsymbol{\mu}, \mathbf{C})$ reduces to $\mathcal{N}_N(\boldsymbol{\mu}, \mathbf{C})$ and q-$\mathcal{EP}(0,\mathcal{C})$ becomes $\mathcal{GP}(0,\mathcal{C})$. When $q \in [1,2)$, q-$\mathcal{EP}(0,\mathcal{C})$ lends flexibility to modeling functional data with more regularization than GP. In practice, $q = 1$ is often adopted for faster posterior convergence (Agapiou et al., 2021; Lan et al., 2023) and the capability of preserving inhomogeneous features (rough functional data, edges in image, etc). Refer to Figure 1 for the regularization effect of $q$.*

One caveat of Q-EP is that uncorrelation (identity covariance) does not imply independence except for the special Gaussian case ($q = 2$). For multiple Q-EPs, $(u_1(x), \cdots, u_D(x))$, we usually do not assume they are independent because their joint distribution is difficult to work with (due to the lack of additivity in the exponential part of density function (1)). Rather, uncorrelation is a preferable assumption. In general, we define multi-output (multivariate) Q-EPs through matrix vectorization.

**Definition 5** *A multi-output (multivariate) q-exponential process, $u(\cdot) = (u_1(\cdot), \cdots, u_D(\cdot))$, each $u_j(\cdot) \sim$ q-$\mathcal{EP}(\mu_j, \mathcal{C}_x)$, is said to have association $\mathbf{C}_t$ if at any finite locations, $\mathbf{x} = \{x_n\}_{n=1}^N$, $\mathrm{vec}([u_1(\mathbf{x}), \cdots, u_D(\mathbf{x})]_{N \times D}) \sim$ q-ED$_{ND}(\mathrm{vec}(\boldsymbol{\mu}), \mathbf{C}_t \otimes \mathbf{C}_x)$, where we have $u_j(\mathbf{x}) = [u_j(x_1), \cdots, u_j(x_N)]^{\mathsf{T}}$, $j = 1, \ldots, D$, $\boldsymbol{\mu} = [\mu_1(\mathbf{x}), \cdots, \mu_D(\mathbf{x})]_{N \times D}$ and $\mathbf{C}_x = [\mathcal{C}_x(x_n, x_m)]_{N \times N}$. We denote $u \sim$ q-$\mathcal{EP}(\mu, \mathcal{C}_x, \mathbf{C}_t)$. In particular, $\{u_j(\cdot)\}$ are m.i.u. if $\mathbf{C}_t = \mathbf{I}_D$.*

To improve the modeling expressiveness of Q-EP, we stack m.i.u. multi-output Q-EPs to build a deep Q-EP, similarly as constructing deep GP with multiple GP layers. For this purpose, we first introduce Bayesian (multivariate) regression with Q-EP priors.

### 2.3. Bayesian Regression with Q-EP Priors

Given data $\mathbf{x} = \{x_n\}_{n=1}^N$ and $\mathbf{y} = \{y_n\}_{n=1}^N$, we consider the generic Bayesian regression model:

$$
\begin{aligned}
\mathbf{y} &= f(\mathbf{x}) + \boldsymbol{\varepsilon}, \quad \boldsymbol{\varepsilon} \sim \text{q-ED}_N(0, \Sigma), \\
f &\sim \text{q-}\mathcal{EP}(0, \mathcal{C}).
\end{aligned}
\tag{2}
$$

It is proved in Theorem 3.5 of Li et al. (2023) that the posterior (predictive) distribution is analytically tractable when both the prior and the likelihood are Q-EP, which is one of the keys for the deep generalization of Q-EP.

**Theorem 6** *For the regression model (2), the posterior distribution of $f(x_*)$ at $x_*$ is*

$$
\begin{aligned}
f(x_*) | \mathbf{y}, \mathbf{x}, x_* &\sim \text{q-ED}(\boldsymbol{\mu}^*, \mathbf{C}^*), \\
\boldsymbol{\mu}^* = \mathbf{C}_*^{\mathsf{T}}(\mathbf{C} + \Sigma)^{-1}\mathbf{y}, \ \mathbf{C}^* &= \mathbf{C}_{**} - \mathbf{C}_*^{\mathsf{T}}(\mathbf{C} + \Sigma)^{-1}\mathbf{C}_*,
\end{aligned}
$$

*where $\mathbf{C} = \mathcal{C}(\mathbf{x}, \mathbf{x})$, $\mathbf{C}_* = \mathcal{C}(\mathbf{x}, x_*)$, and $\mathbf{C}_{**} = \mathcal{C}(x_*, x_*)$.*

Denote $\mathbf{X} = [\mathbf{x}_1, \cdots, \mathbf{x}_Q]_{N \times Q}$, $\mathbf{F} = [f_1(\mathbf{X}), \cdots, f_D(\mathbf{X})]_{N \times D}$ and $\mathbf{Y} = [\mathbf{y}_1, \cdots, \mathbf{y}_D]_{N \times D}$. With m.i.u. Q-EP priors as in Definition (5) imposed on $f := (f_1, \cdots, f_D)$, we now consider the following multivariate regression problem:

$$
\begin{aligned}
\text{likelihood:} \quad & \text{vec}(\mathbf{Y})|\mathbf{F} \sim \text{q-ED}_{ND}(\text{vec}(\mathbf{F}), \mathbf{I}_D \otimes \Sigma), \\
\text{prior on latent function:} \quad & f \sim \text{q-}\mathcal{EP}(0, \mathcal{C}, \mathbf{I}_D).
\end{aligned}
\tag{3}
$$

Based on the additivity of q-ED random variables (Fang and Zhang, 1990), we can find the marginal of $\mathbf{Y}$ by noticing that $\mathbf{Y} = \mathbf{F} + \boldsymbol{\varepsilon}$ with $\text{vec}(\boldsymbol{\varepsilon}) \sim \text{q-ED}(\mathbf{0}, \mathbf{I}_D \otimes \Sigma)$:

$$
\text{marginal likelihood:} \quad \text{vec}(\mathbf{Y})|\mathbf{X} \sim \text{q-ED}_{ND}(\mathbf{0}, \mathbf{I}_D \otimes (\mathbf{C} + \Sigma)).
\tag{4}
$$

## 3. Shallow Q-EP Model

In this section we introduce the shallow (1-layer) Q-EP model which serves as a building block for the deep Q-EP model to be developed in Section 4. We start with the marginal model (4) that can be identified as a latent variable model (LVM) (Lawrence, 2003) with a specified kernel. This defines a shallow Q-EP model. Then we develop variational inference with sparse approximation for such model (Titsias and Lawrence, 2010) and stack multiple layers to build the deep Q-EP.

The marginal model (4) of $\mathbf{Y}|\mathbf{X}$ can be viewed as a stochastic mapping (Theorem 2.1 and Proposition A.1 of Li et al., 2023): $\tilde{f} : \mathbf{X} \to \mathbf{Y} = R\mathbf{L}_\mathbf{X}\mathbf{S}$ , where $R^q \sim \chi^2(N)$, $\mathbf{L}_\mathbf{X}$ is the Cholesky factor of $\mathbf{C}_\mathbf{X} + \Sigma$ whose value depends on $\mathbf{X}$, and $\mathbf{S} := [S_1, \cdots, S_D] \sim \text{Unif}(\prod_{d=1}^D \mathcal{S}^{N+1})$, i.e. each $S_d$ is uniformly distributed on a unit-sphere $\mathcal{S}^{N+1}$.

Note that $\mathbf{X}$ is an input variable in supervised learning, and could also be a latent variable in unsupervised learning. In the latter case, the shallow Q-EP model (4) of $\mathbf{Y}|\mathbf{X}$ can be regarded as an LVM obtained by integrating out the latent function $\mathbf{F}$ in model (3), which is a linear mapping in probabilistic PCA (Tipping and Bishop, 1999) and a multi-output GP in GP-LVM (Lawrence, 2003, 2005). Hence, we propose the shallow Q-EP model as a Q-EP LVM by replacing GP with Q-EP in the LVM.

For the convenience of exposition, we set $\Sigma = \beta^{-1}\mathbf{I}_N$ and denote $\mathbf{K} := \mathbf{C}_\mathbf{X} + \Sigma$. For $\mathbf{K} = [k(\mathbf{x}_n, \mathbf{x}_m)]_{N \times N}$ we adopt the following automatic relevance determination (ARD) kernel as in Titsias and Lawrence (2010), e.g., squared exponential (SE), to determine the dominant dimensions in the input (latent) space:

$$
k(\mathbf{x}_n, \mathbf{x}_m) = \frac{1}{\alpha} \exp \left\{ -\frac{1}{2}(\mathbf{x}_n - \mathbf{x}_m)^\mathsf{T} \text{diag}(\boldsymbol{\gamma})(\mathbf{x}_n - \mathbf{x}_m) \right\}.
\tag{5}
$$

### 3.1. Bayesian Shallow Q-EP

Like Titsias and Lawrence (2010), we adopt a prior for the input (latent) variable $\mathbf{X}$ and introduce the following Bayesian shallow Q-EP model:

$$
\begin{aligned}
\text{marginal likelihood:} \quad & \text{vec}(\mathbf{Y})|\mathbf{X} \sim \text{q-ED}(\mathbf{0}, \mathbf{I}_D \otimes \mathbf{K}), \\
\text{prior on input/latent variable:} \quad & \text{vec}(\mathbf{X}) \sim \text{q-ED}(\mathbf{0}, \mathbf{I}_{NQ}).
\end{aligned}
\tag{6}
$$

Compared with the optimization method (Lawrence, 2003), the Bayesian training procedure is robust to overfitting and can automatically determine the intrinsic dimensionality of the

nonlinear input (latent) space (Titsias and Lawrence, 2010) by thresholding the correlation length-scale $\boldsymbol{\gamma}$.

For more practical applications, we use variational Bayes, instead of Markov Chain Monte Carlo (MCMC), to train the shallow Q-EP model (6). The variational inference for this model is much more complicated than GP-LVM because the log-likelihood (3) is no longer represented as a quadratic form of data. It should be noted that many expectations in the evidence lower bound (ELBO) are no longer analytically tractable with a general power $q$ in the exponent of the density (1), which makes it much more challenging to derive a computable ELBO. We solve this issue with the help of Jensen's inequality.

For variational Bayes, we approximate the posterior distribution $p(\mathbf{X}|\mathbf{Y}) \propto p(\mathbf{Y}|\mathbf{X})p(\mathbf{X})$ with the uncorrelated q-ED: $q(\mathbf{X}) \sim$ q-ED$(\boldsymbol{\mu}, \text{diag}(\{\mathbf{S}_n\}))$, where each covariance $\mathbf{S}_n$ is of size $Q \times Q$ and can be chosen as a diagonal matrix for convenience. To speed up the computation, sparse variational approximation (Titsias, 2009; Lawrence and Moore, 2007) is adopted by introducing the inducing points $\tilde{\mathbf{X}} \in \mathbb{R}^{M \times Q}$ with their function values $\mathbf{U} = [f_1(\tilde{\mathbf{X}}), \cdots, f_D(\tilde{\mathbf{X}})] \in \mathbb{R}^{M \times D}$. Hence the marginal likelihood $p(\mathbf{Y}|\mathbf{X})$ in (6) can be augmented to a joint distribution of several q-ED random variables: $p(\mathbf{Y}|\mathbf{X}) \propto p(\mathbf{Y}|\mathbf{F})p(\mathbf{F}|\mathbf{U}, \mathbf{X}, \tilde{\mathbf{X}})p(\mathbf{U}|\tilde{\mathbf{X}})$, where $p(\text{vec}(\mathbf{F})|\mathbf{U}, \mathbf{X}, \tilde{\mathbf{X}}) \sim$ q-ED$(\text{vec}(\mathbf{K}_{NM}\mathbf{K}_{MM}^{-1}\mathbf{U}), \mathbf{I}_D \otimes (\mathbf{K}_{NN} - \mathbf{K}_{NM}\mathbf{K}_{MM}^{-1}\mathbf{K}_{MN}))$ and $p(\text{vec}(\mathbf{U})|\tilde{\mathbf{X}}) \sim$ q-ED$(\mathbf{0}, \mathbf{I}_D \otimes \mathbf{K}_{MM})$.

Denote $\varphi(r; \Sigma, D) := -\frac{D}{2}\log|\Sigma| + \frac{ND}{2}\left(\frac{q}{2} - 1\right)\log r - \frac{1}{2}r^{\frac{q}{2}}$. With the variational distribution $q(\mathbf{F}, \mathbf{U}, \mathbf{X}) = p(\mathbf{F}|\mathbf{U}, \mathbf{X})q(\mathbf{U})q(\mathbf{X})$ for $q(\mathbf{U}) \sim$ q-ED$(\mathbf{M}, \text{diag}(\{\boldsymbol{\Sigma}_d\}))$, the following final ELBO is obtained by the similar approach in (SVGP Hensman et al., 2015) (Refer to Section A.1 for details):

$$
\begin{aligned}
\log p(\mathbf{Y}) \geq &\mathcal{L}(q) = \int q(\mathbf{X})q(\mathbf{U})p(\mathbf{F}|\mathbf{U}, \mathbf{X}) \log \frac{p(\mathbf{Y}|\mathbf{F})p(\mathbf{U})p(\mathbf{X})}{q(\mathbf{U})q(\mathbf{X})} d\mathbf{F}d\mathbf{U}d\mathbf{X} \\
\geq &h^*(\mathbf{Y}, \mathbf{X}) - \text{KL}_{\mathbf{U}}^* - \text{KL}_{\mathbf{X}}^*, \\
h^*(\mathbf{Y}, \mathbf{X}) = &\varphi(r_{\mathbf{Y}}; \beta^{-1}\mathbf{I}_N, D), \\
r_{\mathbf{Y}} = &r(\mathbf{Y}, \Psi_1 \mathbf{K}_{MM}^{-1}\mathbf{M}) + \beta\text{tr}(\mathbf{M}^{\mathsf{T}}\mathbf{K}_{MM}^{-1}(\Psi_2 - \Psi_1^{\mathsf{T}}\Psi_1)\mathbf{K}_{MM}^{-1}\mathbf{M}) \\
& + \beta D[\psi_0 - \text{tr}(\mathbf{K}_{MM}^{-1}\Psi_2)] + \beta \sum_{d=1}^{D} \text{tr}(\mathbf{K}_{MM}^{-1}\boldsymbol{\Sigma}_d\mathbf{K}_{MM}^{-1}\Psi_2), \qquad (7) \\
-\text{KL}_{\mathbf{U}}^* = &\frac{1}{2}\sum_{d=1}^{D} \log|\boldsymbol{\Sigma}_d| + \varphi\left(\text{tr}(\mathbf{M}^{\mathsf{T}}\mathbf{K}_{MM}^{-1}\mathbf{M}) + \sum_{d=1}^{D}\text{tr}(\boldsymbol{\Sigma}_d\mathbf{K}_{MM}^{-1}); \mathbf{K}_{MM}, D\right), \\
-\text{KL}_{\mathbf{X}}^* = &\frac{1}{2}\sum_{n=1}^{N} \log|\mathbf{S}_n| + \varphi\left(\text{tr}(\boldsymbol{\mu}^{\mathsf{T}}\boldsymbol{\mu}) + \sum_{n=1}^{N}\text{tr}(\mathbf{S}_n); \mathbf{I}_N, Q\right),
\end{aligned}
$$

where $\psi_0 = \text{tr}(\langle\mathbf{K}_{NN}\rangle_{q(\mathbf{X})})$, $\Psi_1 = \langle\mathbf{K}_{NM}\rangle_{q(\mathbf{X})}$, and $\Psi_2 = \langle\mathbf{K}_{MN}\mathbf{K}_{NM}\rangle_{q(\mathbf{X})}$.

**Remark 7** *When $q = 2$, $\varphi(r; \Sigma, D) = -\frac{D}{2}\log|\Sigma| - \frac{1}{2}r$ with $r = r(\mathbf{Y}, \Psi_1\mathbf{K}_{MM}^{-1}\mathbf{M})$ becomes the log-density of matrix normal $\mathcal{MN}_{N \times D}(\Psi_1\mathbf{K}_{MM}^{-1}\mathbf{M}, \beta^{-1}\mathbf{I}_N, \mathbf{I}_D)$. Then the ELBO (7) reduces to the ELBO as in Equation (7) of (SVGP Hensman et al., 2015) with an extra term $\beta\text{tr}(\mathbf{M}^{\mathsf{T}}\mathbf{K}_{MM}^{-1}(\Psi_2 - \Psi_1^{\mathsf{T}}\Psi_1)\mathbf{K}_{MM}^{-1}\mathbf{M})$. The computational complexity, $\mathcal{O}(NM^2)$, remains the same as GP-LVM (Titsias and Lawrence, 2010).*

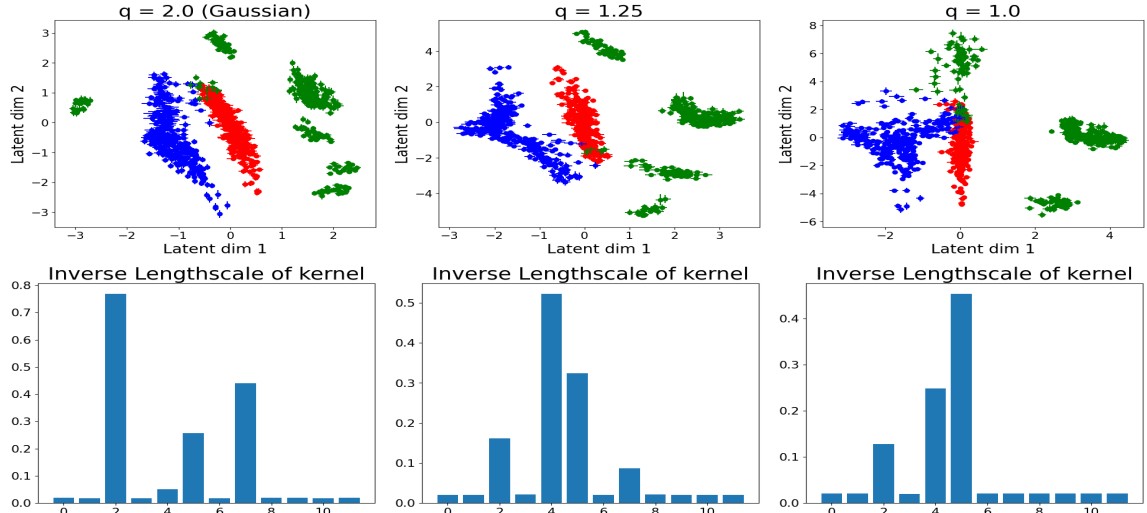

Figure 1: 2d latent space of multi-phase oil-flow dataset: contrasting GP-LVM ($q = 2$) (left) with two shallow Q-EPs for $q = 1.25$ (middle) and $q = 1$ (right). Smaller $q$ tends to contract the latent space and hence regularizes the learned latent representation, an effect similarly existing among ridge regression, elastic-net, and Lasso.

We demonstrate the behavior of shallow Q-EP as an LVM in unsupervised learning and contrast it with GP-LVM using the canonical multi-phase oil-flow dataset (Titsias and Lawrence, 2010) that consists of 1000 observations (12-dimensional) corresponding to three different phases of oil-flow. Figure 1 visualizes the 2d latent subspaces identified with two most dominant latent dimensions found by GP-LVM (left) and two shallow Q-EP models with $q = 1.25$ (middle) and $q = 1$ (right) respectively. The vertical and horizontal bars indicate axis aligned uncertainty around each latent point. As GP-LVM corresponds to a shallow Q-EP with $q = 2$, the parameter $q > 0$ controls a regularization effect of shallow Q-EP: the smaller $q$ leads to more regularization on the learned latent representations and hence yields clusters more aggregated, as illustrated by the green class in upper row of Figure 1. All models identify three intrinsic dimensions, as indicated by three dominant inverse lengthscales in the lower row.

## 4. Deep Q-EP Model

Now we construct the deep Q-EP model by stacking multiple shallow Q-EP layers introduced in Section 3, similarly as building deep GP with GP-LVMs (Damianou and Lawrence, 2013). More specifically, we consider a hierarchy of $L$ shallow Q-EP layers (6) as follows:

$$
\begin{aligned}
y_{nd} &= f_d^0(\mathbf{x}_n^1) + \varepsilon_{nd}^0, \quad d = 1, \cdots, D_0, \quad \mathbf{x}_n^1 \in \mathbb{R}^{D_1}, \\
x_{nd}^1 &= f_d^1(\mathbf{x}_n^2) + \varepsilon_{nd}^1, \quad d = 1, \cdots, D_1, \quad \mathbf{x}_n^2 \in \mathbb{R}^{D_2}, \\
&\vdots \qquad\qquad \vdots \qquad\qquad \vdots \qquad\qquad \vdots \\
x_{nd}^{L-1} &= f_d^{L-1}(\mathbf{z}_n) + \varepsilon_{nd}^{L-1}, \ d = 1, \cdots, D_{L-1}, \ \mathbf{z}_n \in \mathbb{R}^{D_L},
\end{aligned}
$$

where $\boldsymbol{\varepsilon}^\ell \sim$ q-ED$(\mathbf{0}, \Gamma^\ell)$, $f^\ell \sim$ q-$\mathcal{EP}(0, k^\ell, I_{D_\ell})$ for $\ell = 0, \cdots, L-1$ and $\mathbf{Y} = \mathbf{X}^0$, $\mathbf{Z} = \mathbf{X}^L$.

Consider the prior $\mathbf{Z} \sim$ q-ED$(\mathbf{0}, \mathbf{I}_{ND_L})$. The joint probability, augmented with the inducing points $\tilde{\mathbf{X}}^{\ell+1}$ and the associated function values $\mathbf{U}^\ell = [f_d^\ell(\tilde{\mathbf{X}}^{\ell+1})]_{d=1}^{D_{\ell+1}}$, is decomposed as $p(\{\mathbf{X}^\ell, \mathbf{F}^\ell, \mathbf{U}^\ell\}_{\ell=0}^{L-1}, \mathbf{Z}) = \prod_{\ell=0}^{L-1} p(\mathbf{X}^\ell | \mathbf{F}^\ell) p(\mathbf{F}^\ell | \mathbf{U}^\ell, \mathbf{X}^{\ell+1}) p(\mathbf{U}^\ell) \cdot p(\mathbf{Z})$. And we use the variational distribution $\mathcal{Q} = \prod_{\ell=0}^{L-1} p(\mathbf{F}^\ell | \mathbf{U}^\ell, \mathbf{X}^{\ell+1}) q(\mathbf{U}^\ell) q(\mathbf{X}^{\ell+1})$, with $q(\mathbf{X}^{\ell+1}) =$ q-ED$(\boldsymbol{\mu}^{\ell+1}, \mathrm{diag}(\{\mathbf{S}_n^{\ell+1}\}))$. Then the ELBO becomes

$$\mathcal{L}(\mathcal{Q}) = \int_{\{\mathbf{F}^\ell, \mathbf{U}^\ell, \mathbf{X}^{\ell+1}\}_{\ell=0}^{L-1}} \mathcal{Q} \log \frac{p(\{\mathbf{X}^\ell, \mathbf{F}^\ell, \mathbf{U}^\ell\}_{\ell=0}^{L-1}, \mathbf{Z})}{\prod_{\ell=0}^{L-1} q(\mathbf{U}^\ell) q(\mathbf{X}^{\ell+1})} \prod_{\ell=0}^{L-1} d\mathbf{F}^\ell d\mathbf{U}^\ell d\mathbf{X}^{\ell+1}$$

$$= h_0 - \mathrm{KL}_{\mathbf{U}^0} + \sum_{\ell=1}^{L-1} [h_\ell - \mathrm{KL}_{\mathbf{U}^\ell} + \mathcal{H}_q(\mathbf{X}_\ell)] - \mathrm{KL}_{\mathbf{Z}},$$

where $h_\ell = \langle \log p(\mathbf{X}^\ell | \mathbf{F}^\ell) \rangle_{q(\mathbf{F}^\ell) q(\mathbf{X}^{\ell+1}) q(\mathbf{X}^\ell)}$ with $q(\mathbf{X}^0) = q(\mathbf{Y}) \equiv 1$. Based on the previous bound (7), we have for $\ell = 1, \cdots, L-1$ (Refer to Section A.2 for details):

$$h_0 \geq h^*(\mathbf{Y}, \mathbf{X}^1),\ h_\ell \geq h^*(\mathbf{X}^\ell, \mathbf{X}^{\ell+1}) = \varphi(r_{\boldsymbol{\mu}^\ell}; \Gamma^\ell, D_\ell),\ -\mathrm{KL}_{\mathbf{U}^\ell} \geq -\mathrm{KL}_{\mathbf{U}^\ell}^*,\ \mathcal{H}_q(\mathbf{X}_\ell) \geq \mathcal{H}_{\mathbf{X}_\ell}^*$$

$$r_{\boldsymbol{\mu}^\ell} = r(\boldsymbol{\mu}^\ell, \Psi_1^\ell (\mathbf{K}_{MM}^\ell)^{-1} \mathbf{M}^\ell) + \mathrm{tr}((\mathbf{M}^\ell)^{\mathsf{T}} (\mathbf{K}_{MM}^\ell)^{-1} (\Psi_2^\ell - (\Psi_1^\ell)^{\mathsf{T}} (\Gamma^\ell)^{-1} \Psi_1^\ell)(\mathbf{K}_{MM}^\ell)^{-1} \mathbf{M}^\ell)$$

$$+ D_\ell [\psi_0^\ell - \mathrm{tr}((\mathbf{K}_{MM}^\ell)^{-1} \Psi_2^\ell)] + \sum_{d=1}^{D_\ell} \mathrm{tr}((\mathbf{K}_{MM}^\ell)^{-1} \boldsymbol{\Sigma}_d^\ell (\mathbf{K}_{MM}^\ell)^{-1} \Psi_2^\ell)$$

$$+ \mathrm{tr}((\mathbf{I}_{D_\ell} \otimes (\Gamma^\ell)^{-1}) \mathrm{diag}(\{\mathbf{S}_n^\ell\})),$$

$$-\mathrm{KL}_{\mathbf{U}^\ell}^* = \frac{1}{2} \sum_{d=1}^{D_\ell} \log |\boldsymbol{\Sigma}_d^\ell| + \varphi \left( \mathrm{tr}((\mathbf{M}^\ell)^{\mathsf{T}} (\mathbf{K}_{MM}^\ell)^{-1} \mathbf{M}^\ell) + \sum_{d=1}^{D_\ell} \mathrm{tr}(\boldsymbol{\Sigma}_d^\ell (\mathbf{K}_{MM}^\ell)^{-1}); \mathbf{K}_{MM}^\ell, D_\ell \right),$$

$$\mathcal{H}_{\mathbf{X}_\ell}^* = \frac{1}{2} \sum_{n=1}^{N} \log |\mathbf{S}_n^\ell|,\ -\mathrm{KL}_{\mathbf{Z}}^* = \frac{1}{2} \sum_{n=1}^{N} \log |\mathbf{S}_n^L| + \varphi \left( \mathrm{tr}((\boldsymbol{\mu}^L)^{\mathsf{T}} \boldsymbol{\mu}^L) + \sum_{n=1}^{N} \mathrm{tr}(\mathbf{S}_n^L); \mathbf{I}_N, D_L \right),$$

where $\psi_0^\ell = \mathrm{tr}((\Gamma^\ell)^{-1} \langle \mathbf{K}_{NN}^\ell \rangle_{q(\mathbf{X}^{\ell+1})})$, $\Psi_1^\ell = \langle \mathbf{K}_{NM}^\ell \rangle_{q(\mathbf{X}^{\ell+1})}$, and $\Psi_2^\ell = \langle \mathbf{K}_{MN}^\ell \mathbf{K}_{NM}^\ell \rangle_{q(\mathbf{X}^{\ell+1})}$.

## 5. Numerical Experiments

In this section, we compare our proposed deep Q-EP with deep GP (DGP Damianou and Lawrence, 2013), deep kernel learning with GP (DKL-GP Wilson et al., 2016), and deep sigma point process (DSPP Jankowiak et al., 2020b) using simulated and benchmark datasets. In simulations, deep Q-EP model manifests unique features in properly modeling inhomogeneous data. For benchmark regression and classification problems, deep Q-EP demonstrates superior or comparable numerical performance. In most cases, 2 layer structure is sufficient for deep Q-EP to have superior or comparable performance compared with deep GP, and DSPP. A large feature extracting neural network (DNN with structure $D_L - 1000 - 500 - 50 - D_0$) is employed before one GP layer for DKL-GP unless stated otherwise. The Matérn kernel ($\nu = 1.5$) is adopted for all the models with trainable hyperparameters (magnitude and correlation strength) and $q = 1$ is chosen in Q-EP and deep Q-EP models for handling data inhomogeneity. All the computer codes are implemented in `GPyTorch` (Gardner et al., 2018) available at `https://github.com/lanzithinking/DeepQEP`.

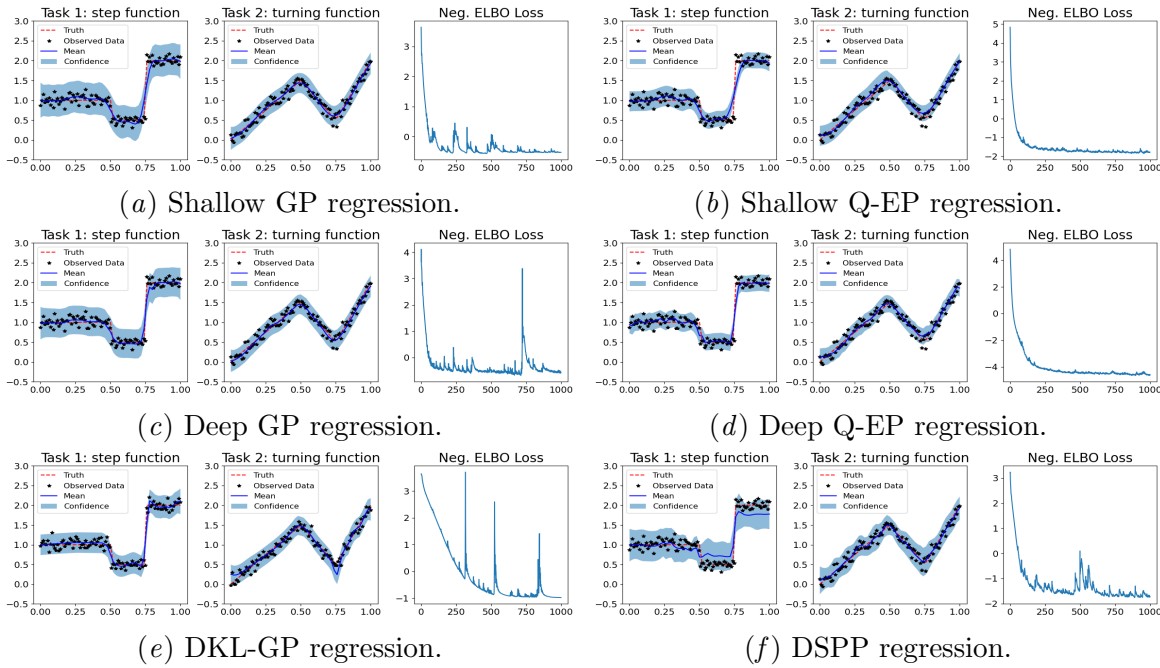

Figure 2: Comparing deep Q-EP (2(*d*)) with cutting-edge deep models including deep GP (2(*c*)), DKL-GP (2(*e*)) and DSPP (2(*f*)) on modeling a 2d-output time series.

## 5.1. Regression

**Time Series** We first consider a simulated 2-dimensional time series from Li et al. (2023), one with step jumps and the other with sharp turnings, whose true trajectories are as follows:

$$u_{\mathrm{J}}(t) = 1, \quad t \in [0,1]; \quad 0.5, \quad t \in (1, 1.5]; \quad 2, \quad t \in (1.5, 2]; \quad 0, \quad \textit{otherwise};$$
$$u_{\mathrm{T}}(t) = 1.5t, \quad t \in [0,1]; \quad 3.5 - 2t, \quad t \in (1, 1.5]; \quad 3t - 4, \quad t \in (1.5, 2]; \quad 0, \quad \textit{otherwise}.$$

We generate time series $\{\mathbf{y}_i\}_{i=1}^{N}$ by adding Gaussian noises to the true trajectories evaluated at $N = 100$ evenly spaced points $t_i \in [0, 2]$, i.e., $\mathbf{y}_i^* = [u_J(t_i), u_T(t_i)]^{\mathsf{T}} + \boldsymbol{\varepsilon}_i, \ \boldsymbol{\varepsilon}_i \overset{iid}{\sim} N(\mathbf{0}, \sigma^2 \mathbf{I}_2), \ with \ \sigma = 0.1, \ i = 1, \cdots, N$. Then we make prediction over 50 points evenly spread over $[0, 2]$.

Abrupt changes exist in these time series for either values or directions, hence pose challenges for standard GP as an $L_2$ penalty based regression method. As shown in Figure 2, results by both deep GP and deep Q-EP are comparatively better than their shallow (one-layer) versions. Among these models, deep Q-EP yields the most accurate prediction and the tightest uncertainty bound (refer to Table B.1) due to its $L_1$ regularization feature that is more suitable to capture these abrupt changes. The loss of (deep) Q-EP model may not be comparable to those for other models because they are based on different probability distributions, and yet it converges faster and and more stably than GP (and the other two benchmark deep probabilistic models), supporting its advantage in convergence (Remark 4). Both DKL-GP and DSPP suffer from slow convergence and unstable training. As seen in Table B.1 comparing mean of absolute error (MAE), standard deviation (STD) of

variational distribution and coefficient of determination ($R^2$), their results possess larger standard errors from repeated experiments, even though few individual runs may yield better results than Deep Q-EP.

**UCI Regression Benchmark**  Next, we test deep Q-EP on a series of benchmark regression datasets (Wilson et al., 2016; Jankowiak et al., 2020b) from UCI machine learning repository. They are selected to represent data at different scales. As in Table 1, for most cases, deep Q-EP demonstrates superior or comparable performance measured by testing data in terms of MAE (accuracy), STD (uncertainty) and NLL because the Q-EP prior provides crucial regularization for datasets where sparse regression is more appropriate. Note that, the marginal likelihood (NLL) values are not comparable among different models (with distinct probability distributions, refer to (1)) and are only listed for reference. As the data volume increases, DNN feature extractor starts to catch up so that DKL-GP surpasses the vanilla deep Q-EP in the song dataset.

Table 1: Regression on UCI datasets: mean of absolute error (MAE), standard deviation of predictive distribution (PSD) and negative logarithm of marginal likelihood (NLL) values by various deep models. Each result of the upper part is averaged over 10 experiments with different random seeds; values in the lower part are standard errors of these repeated experiments.

| Dataset | N, d | Deep GP | | | Deep Q-EP | | | DKL-GP | | | DSPP | | |
|---|---|---|---|---|---|---|---|---|---|---|---|---|---|
| | | MAE | PSD | NLL | MAE | PSD | NLL | MAE | PSD | NLL | MAE | PSD | NLL |
| concrete | 1030, 8 | 10.586 | **1.846** | 25.473 | **9.114** | 2.179 | 4.020 | 9.770 | 2.943 | 10.837 | 10.740 | 2.567 | 9.882 |
| gas | 2565, 128 | 0.187 | 0.395 | 0.402 | **0.136** | **0.163** | 1.069 | 0.965 | 0.611 | 2.236 | 0.292 | 0.385 | -0.431 |
| elevators | 16599, 18 | 0.0639 | 0.088 | -1.035 | **0.0636** | **0.067** | -0.008 | 0.101 | 0.084 | -0.197 | 0.066 | 0.087 | -1.005 |
| protein | 45730, 9 | 0.385 | 0.526 | 0.755 | **0.351** | 0.363 | 1.873 | 0.364 | 0.425 | 0.769 | 0.365 | **0.208** | 0.148 |
| song | 515345, 90 | 0.379 | 0.478 | 0.686 | 0.398 | 0.397 | 1.869 | **0.355** | 0.440 | 0.640 | 0.394 | 0.235 | 0.501 |
| concrete | 1030, 8 | 0.681 | 0.010 | 2.644 | 0.809 | 0.032 | 0.054 | 0.504 | 0.113 | 0.946 | 1.675 | 0.506 | 2.312 |
| gas | 2565, 128 | 0.071 | 0.058 | 0.162 | 0.027 | 0.027 | 0.107 | 0.291 | 0.082 | 0.755 | 0.239 | 0.241 | 0.874 |
| elevators | 16599, 18 | 2.86e-4 | 4.08e-4 | 6.79e-3 | 3.95e-4 | 1.97e-4 | 6.24e-3 | 0.070 | 5.97e-3 | 1.738 | 9.70e-4 | 0.024 | 0.056 |
| protein | 45730, 9 | 4.77e-3 | 4.16e-3 | 7.47e-3 | 4.46e-3 | 3.83e-3 | 0.011 | 0.083 | 0.040 | 0.197 | 8.84e-3 | 0.024 | 0.016 |
| song | 515345, 90 | 1.73e-3 | 4.13e-3 | 3.98e-3 | 0.041 | 0.039 | 0.084 | 5.48e-3 | 5.08e-3 | 0.016 | 0.029 | 0.124 | 0.191 |

## 5.2. Classification

**Simulation with Inhomogeneous Boundaries**  Consider a simulated classification problem with labels created on annular regions of a rhombus for $i = 1, \cdots, N$:

$$y_i = [\cos(0.4 * u * \pi \|\mathbf{x}_i\|_1)] + 1, \quad u \sim \text{Unif}[0,1], \quad \mathbf{x}_i \sim \mathcal{N}(\mathbf{0}, \mathbf{I}_2),$$

where $[x]$ rounds $x$ to the nearest integer. We generate $N = 500$ random data points according to the formula which results in 3 classes' labels as illustrated in the leftmost panel of Figure 3. Note, the class regions have clear shapes with edges and are not simply connected. Q-EP and deep Q-EP are superior than their GP rivals in modeling such inhomogeneous data. Indeed, Figure 3 shows that even with small amount of data, Q-EP has better decision boundaries than GP and a 3-layer deeper Q-EP yields the best result closest to the truth among all the models. On the contrary, (deep) GP tends to yield round and over-smooth decision boundaries because of its $L_2$ nature. This is further illustrated in Figure B.1 with more fine details revealed by the logits. Note, it is understandable that none of these models characterizes the correct boundary around the corners due to the absence of data. Table B.2 compares their performance on testing data in terms of classification accuracy (ACC), area under ROC curve (AUC) and deep Q-EP achieves the highest accuracy.

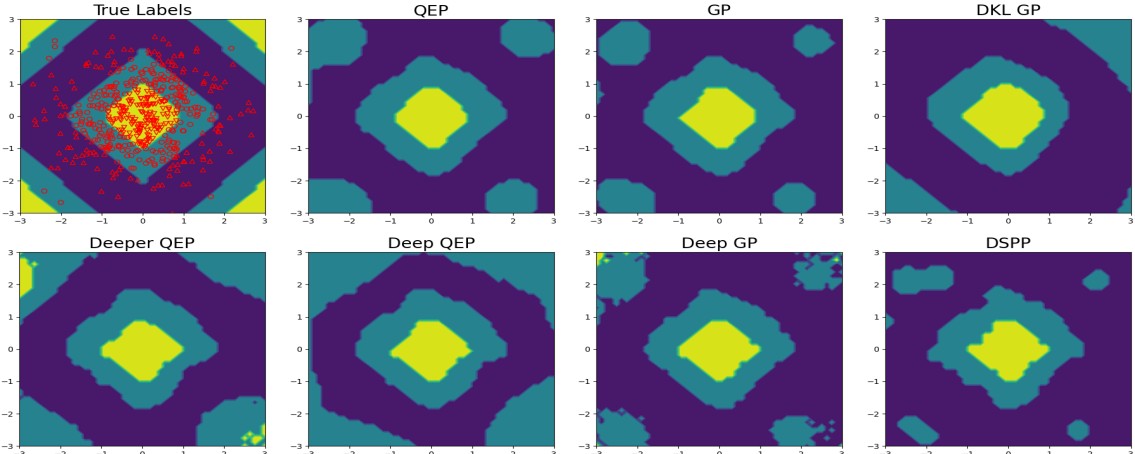

Figure 3: Comparing shallow (1-layer), deep (2-layer) and deeper (3-layer) Q-EPs with GP, deep GP, DKL-GP and DSPP on a classification problem defined on annular rhombus. Circles, upper and lower triangles label three classes in the training data.

Table 2: Classification on UCI datasets: accuracy (ACC), area under ROC curve (AUC) and negative logarithm of marginal likelihood (NLL) values by various deep models. Each result of the upper part is averaged over 10 experiments with different random seeds; values in the lower part are standard errors of these repeated experiments.

| Dataset | N, d, k | Deep GP | | | Deep Q-EP | | | DKL-GP | | | DSPP | | |
|---|---|---|---|---|---|---|---|---|---|---|---|---|---|
| | | ACC | AUC | NLL | ACC | AUC | NLL | ACC | AUC | NLL | ACC | AUC | NLL |
| haberman | 306, 3, 2 | 0.727 | 0.460 | 5.803 | **0.732** | **0.505** | 5.516 | 0.690 | 0.511 | 6.060 | 0.716 | 0.496 | 27.046 |
| dermatology | 366, 94, 4 | 0.443 | 0.494 | 14.972 | **0.512** | **0.527** | 14.330 | 0.339 | 0.515 | 16.508 | 0.458 | 0.482 | 72.449 |
| tic-tac-toe | 957, 27, 2 | 0.971 | 0.515 | 2.719 | **0.972** | 0.532 | -0.572 | 0.885 | **0.653** | 3.692 | 0.736 | 0.503 | 233.559 |
| car | 1728, 21, 4 | **0.990** | **1.000** | 2.237 | 0.983 | 0.999 | -0.630 | 0.737 | 0.826 | 8.906 | 0.758 | 0.848 | 7.48e3 |
| nursery | 12959, 27, 5 | **0.9996** | 0.967 | 2.841 | **0.9996** | **0.982** | -13.401 | 0.503 | 0.654 | 69.032 | 0.717 | 0.839 | 1.43e4 |
| haberman | 306, 3, 2 | 0.012 | 0.081 | 0.271 | 0.020 | 0.072 | 1.156 | 0.092 | 0.093 | 0.728 | 0.031 | 0.048 | 42.901 |
| dermatology | 366, 94, 4 | 0.053 | 0.047 | 1.303 | 0.071 | 0.054 | 0.510 | 0.102 | 0.052 | 1.505 | 0.072 | 0.048 | 8.668 |
| tic-tac-toe | 957, 27, 2 | 0.021 | 0.081 | 0.407 | 0.040 | 0.369 | 0.880 | 0.270 | 0.172 | 2.271 | 0.227 | 0.442 | 123.140 |
| car | 1728, 21, 4 | 9.14e-3 | 2.27e-4 | 1.208 | 7.04e-3 | 1.34e-3 | 1.927 | 0.378 | 0.282 | 4.586 | 0.220 | 0.183 | 1.03e4 |
| nursery | 12959, 27, 5 | 6.28e-8 | 0.042 | 4.664 | 6.28e-8 | 0.031 | 94.847 | 0.416 | 0.275 | 78.647 | 0.178 | 0.083 | 2.01e4 |

**UCI Classification Benchmark**   We also compare deep Q-EP with other deep probabilistic models on several benchmark classification datasets with different sizes from UCI machine learning repository. Table 2 summarizes the comparison results in terms of ACC, AUC and NLL. Deep Q-EP still excels in most cases or has comparable performance, further supporting its advantage in the classification task.

### 5.3. Inverse Reconstruction of Tomography Image

Computed tomography (CT) is a medical imaging technique to obtain internal details of human body by measuring X-ray signals through body tissues at multiple angles. The internal image, $\mathbf{X}$, viewed as a function of pixels on the unit square $[0, 1]^2$ discretized with size $n \times n$, is mapped by a known Radon transformation, $T$, to obtain the noise ($\boldsymbol{\varepsilon}$) contaminated observations (known as sinogram) $\mathbf{Y}$ with $n_s$ sensors at $n_\theta$ angles:

$$\text{vec}(\mathbf{Y}) = T\text{vec}(\mathbf{X}) + \boldsymbol{\varepsilon}, \quad \mathbf{Y} \in \mathbb{R}^{n_\theta \times n_s}, \quad T \in \mathbb{R}^{n_\theta n_s \times n^2}, \quad \mathbf{X} \in \mathbb{R}^{n \times n}.$$

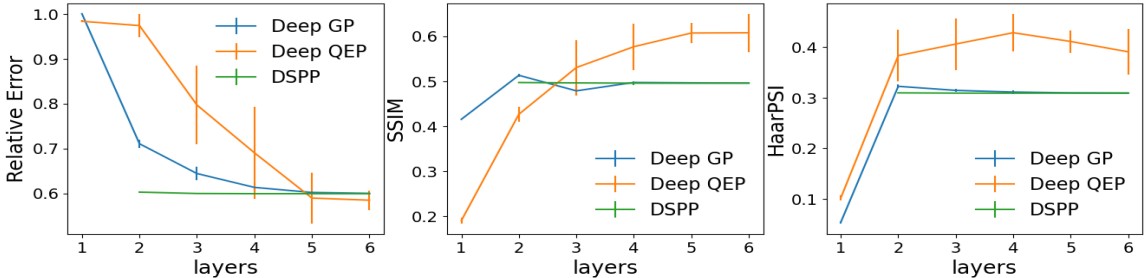

Figure 4: Shepp-Logan phantom: relative error, SSIM, and HarrPSI of reconstructed images obtained by deep GP, deep QEP and DSPP respectively with various depths. Each configuration is repeated for 10 times with different random seeds and the standard errors for the repeated results are indicated by the error bars.

We consider the Shepp-Logan phantom (Shepp and Logan, 1974) as an inverse problem for reconstructing the true CT image, $\mathbf{X}^\dagger$, from the observed sinogram, $\mathbf{Y}$, using deep probabilistic models. In this example, we set $n = 128$, $n_\theta = 90$, $n_s = 100$ and generate $\boldsymbol{\varepsilon}$ with signal noise ratio $\|T\mathbf{X}^\dagger\|/\|\boldsymbol{\varepsilon}\| = 100$. The high-dimensional ($Q = 128^2 = 16,384$) true image ($\mathbf{X}^\dagger$) is illustrated in the leftmost of Figure B.4, which also compares different reconstructions, $\hat{\mathbf{X}}$, by deep GP, deep QEP and DSPP. We omit DKL-GP due to its incomparably worse results. In Figure 4, we further compare these models with various depths in terms of the relative error ($\|\hat{\mathbf{X}} - \mathbf{X}^\dagger\|/\|\mathbf{X}^\dagger\|$), the structured similarity index (SSIM) (Wang et al., 2004), and the Haar wavelet-based perceptual similarity index (HaarPSI) (Reisenhofer et al., 2018). We observe that the model performance (lower errors and higher imaging metrics) improves while the number of layers (depth) increases. The deep QEP outperforms the other two with sufficient depths (5 in relative error, 3 in SSIM and all cases in HaarPSI).

## 6. Conclusion

In this paper, we generalize Q-EP to deep Q-EP, which includes deep GP as a special case. Moreover, deep Q-EP inherits the flexible regularization controlled by a parameter $q > 0$, which is advantageous in learning latent representations and modeling data inhomogeneity. We first generalize Bayesian GP-LVM to Bayesian QEP-LVM (as shallow Q-EP layer) and develop the variational inference for it. Then we stack multiple shallow Q-EP layers to build the deep Q-EP model. The novel deep model demonstrates numerical benefits in various learning tasks and can be combined with neural network for better characterizing complex latent representations in different data applications.

As common in GP and NN models, we do observe multi-modality of the posterior distributions, especially in the hyper-parameter spaces. Sub-optimal solutions can appear in the stochastic training process. These issues can be alleviated by dispersed or diversified initialization, or with adaptive training schedulers. One potential application of deep Q-EP is the inverse learning, similarly as done by deep GP (Jin et al., 2017; Abraham and Deo, 2023). Theory of the contraction properties (Finocchio and Schmidt-Hieber, 2023) is also an interesting research direction.

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

# Supplement Document for "Deep Q-Exponential Processes"

## Appendix A. Computation of Variational Lower Bounds

### A.1. Shallow Q-EP

The variational lower bound for the log-evidence is

$$\log p(\mathbf{Y}) \geq \mathcal{L}(q) := \int q(\mathbf{X}) \log \frac{p(\mathbf{Y}|\mathbf{X})p(\mathbf{X})}{q(\mathbf{X})} d\mathbf{X} = \tilde{\mathcal{L}}(q) - \mathrm{KL}(q(\mathbf{X})\|p(\mathbf{X})),$$

where the first term $\tilde{\mathcal{L}}(q) = \int q(\mathbf{X}) \log p(\mathbf{Y}|\mathbf{X}) d\mathbf{X}$ is intractable and hence difficult to bound.

#### A.1.1. LOWER BOUND FOR THE MARGINAL LIKELIHOOD

To address such intractability issue and speed up the computation, sparse variational approximation (Titsias, 2009; Lawrence and Moore, 2007) is adopted by introducing a set of inducing points $\tilde{\mathbf{X}} \in \mathbb{R}^{M \times Q}$ with their function values $\mathbf{U} = [f_1(\tilde{\mathbf{X}}), \cdots, f_D(\tilde{\mathbf{X}})] \in \mathbb{R}^{M \times D}$. Hence the marginal likelihood $p(\mathbf{Y}|\mathbf{X})$ defined in (6) can be augmented to the following joint distribution each being a q-ED:

$$p(\mathbf{Y}|\mathbf{X}) \propto p(\mathbf{Y}|\mathbf{F})p(\mathbf{F}|\mathbf{U},\mathbf{X},\tilde{\mathbf{X}})p(\mathbf{U}|\tilde{\mathbf{X}}),$$

where we have $\mathrm{vec}(\mathbf{U})|\tilde{\mathbf{X}} \sim \text{q-ED}(\mathbf{0}, \mathbf{I}_D \otimes \mathbf{K}_{MM})$ and the conditional distribution

$$\mathrm{vec}(\mathbf{F})|\mathbf{U},\mathbf{X},\tilde{\mathbf{X}} \sim \text{q-ED}(\mathrm{vec}(\mathbf{K}_{NM}\mathbf{K}_{MM}^{-1}\mathbf{U}), \mathbf{I}_D \otimes (\mathbf{K}_{NN} - \mathbf{K}_{NM}\mathbf{K}_{MM}^{-1}\mathbf{K}_{MN})). \quad (8)$$

The inducing points $\tilde{\mathbf{X}}$ are regarded as variational parameters and hence they are dropped from the following probability expressions. We then approximate $p(\mathbf{F},\mathbf{U}|\mathbf{X}) \propto p(\mathbf{F}|\mathbf{U},\mathbf{X})p(\mathbf{U})$ with $q(\mathbf{F},\mathbf{U}) = p(\mathbf{F}|\mathbf{U},\mathbf{X})q(\mathbf{U})$ in another variational Bayes as follows

$$\begin{aligned}
\log p(\mathbf{Y}|\mathbf{X}) &\geq \int q(\mathbf{F},\mathbf{U}) \log \frac{p(\mathbf{Y}|\mathbf{F})p(\mathbf{F}|\mathbf{U},\mathbf{X})p(\mathbf{U})}{q(\mathbf{F},\mathbf{U})} d\mathbf{F}d\mathbf{U} \\
&= \int p(\mathbf{F}|\mathbf{U})q(\mathbf{U})d\mathbf{U} \log p(\mathbf{Y}|\mathbf{F})d\mathbf{F} + \int q(\mathbf{U}) \log \frac{p(\mathbf{U})}{q(\mathbf{U})} d\mathbf{U}.
\end{aligned} \quad (9)$$

Different from Titsias (2009); Titsias and Lawrence (2010) using the variational calculus, (SVGP Hensman et al., 2015) computes the marginal likelihood ELBO (9) in two stages. Instead of the variational free form, we follow Hensman et al. (2015) to use the variational distribution for $\mathbf{U}$ of the following format conjugate to $p(\mathbf{F}|\mathbf{U})$:

$$q(\mathbf{U}) \sim \text{q-ED}(\mathbf{M}, \mathrm{diag}(\{\mathbf{\Sigma}_d\})). \quad (10)$$

Noticing that $\mathbf{F}|\mathbf{U}$ follows a conditional $q$-exponential (8), we can obtain the variational distribution of $\mathbf{F}$, $q(\mathbf{F})$, by marginalizing $\mathbf{U}$ out as follows

$$\begin{aligned}
q(\mathbf{F}) &= \int q(\mathbf{F},\mathbf{U})d\mathbf{U} = \int p(\mathbf{F}|\mathbf{U})q(\mathbf{U})d\mathbf{U} \\
&\sim \text{q-ED}(\mathrm{vec}(\mathbf{K}_{NM}\mathbf{K}_{MM}^{-1}\mathbf{M}), \\
&\quad \mathbf{I}_D \otimes (\mathbf{K}_{NN} - \mathbf{K}_{NM}\mathbf{K}_{MM}^{-1}\mathbf{K}_{MN}) + \mathrm{diag}(\{\mathbf{K}_{NM}\mathbf{K}_{MM}^{-1}\mathbf{\Sigma}_d\mathbf{K}_{MM}^{-1}\mathbf{K}_{MN}\})).
\end{aligned}$$

Therefore, the variational lower bound of the marginal likelihood (9) becomes

$$\log p(\mathbf{Y}|\mathbf{X}) \geq \langle \log p(\mathbf{Y}|\mathbf{F}) \rangle_{q(\mathbf{F})} - \mathrm{KL}(q(\mathbf{U})\|p(\mathbf{U})).$$

Note, $\log p(\mathbf{Y}|\mathbf{F})$ is not a random quadratic form in general and hence the expectation in the first term has no explicit formula. Denote by $\log p(\mathbf{Y}|\mathbf{F}) = \varphi(r(\mathbf{Y},\mathbf{F}))$, where $\varphi(r) := \frac{DN}{2}\log\beta + \frac{ND}{2}\left(\frac{q}{2}-1\right)\log r - \frac{1}{2}r^{\frac{q}{2}}$ is convex for $q \in (0,2]$, and $r(\mathbf{Y},\mathbf{F}) = \mathrm{vec}(\mathbf{Y}-\mathbf{F})^{\mathsf{T}}(\beta^{-1}\mathbf{I}_{ND})^{-1}\mathrm{vec}(\mathbf{Y}-\mathbf{F}) = \beta\mathrm{tr}((\mathbf{Y}-\mathbf{F})(\mathbf{Y}-\mathbf{F})^{\mathsf{T}})$ is a quadratic form of random variable $\mathbf{Y}$. Therefore, by Jensen's inequality, we can bound from below as

$$\langle \log p(\mathbf{Y}|\mathbf{F}) \rangle_{q(\mathbf{F})} = \langle \varphi(r(\mathbf{Y},\mathbf{F})) \rangle_{q(\mathbf{F})} \geq \varphi(\langle r(\mathbf{Y},\mathbf{F}) \rangle_{q(\mathbf{F})}).$$

where we can calculate the expectation of the quadratic form $r(\mathbf{Y},\mathbf{F})$ as

$$\langle r(\mathbf{Y},\mathbf{F}) \rangle_{q(\mathbf{F})} = r(\mathbf{Y},\mathbf{K}_{NM}\mathbf{K}_{MM}^{-1}\mathbf{M}) + \beta D\mathrm{tr}(\mathbf{K}_{NN} - \mathbf{K}_{NM}\mathbf{K}_{MM}^{-1}\mathbf{K}_{MN})$$
$$+ \beta\sum_{d=1}^{D}\mathrm{tr}(\mathbf{K}_{NM}\mathbf{K}_{MM}^{-1}\boldsymbol{\Sigma}_d\mathbf{K}_{MM}^{-1}\mathbf{K}_{MN}).$$

Denote by $h(\mathbf{Y},\mathbf{X}) = \langle\langle\log p(\mathbf{Y}|\mathbf{F})\rangle_{q(\mathbf{F})}\rangle_{q(\mathbf{X})}$. Then we solve the intractable expectation by another Jensen's inequality

$$h(\mathbf{Y},\mathbf{X}) \geq \varphi(\langle\langle r(\mathbf{Y},\mathbf{F})\rangle_{q(\mathbf{F})}\rangle_{q(\mathbf{X})}) =: h^*(\mathbf{Y},\mathbf{X}).$$

Define $\psi_0 = \mathrm{tr}(\langle\mathbf{K}_{NN}\rangle_{q(\mathbf{X})})$, $\Psi_1 = \langle\mathbf{K}_{NM}\rangle_{q(\mathbf{X})}$, and $\Psi_2 = \langle\mathbf{K}_{MN}\mathbf{K}_{NM}\rangle_{q(\mathbf{X})}$. Further we calculate the expectations of quadratic terms similarly

$$\langle\langle r(\mathbf{Y},\mathbf{F})\rangle_{q(\mathbf{F})}\rangle_{q(\mathbf{X})} = \langle r(\mathbf{Y},\mathbf{K}_{NM}\mathbf{K}_{MM}^{-1}\mathbf{M})\rangle_{q(\mathbf{X})} + \beta D[\psi_0 - \mathrm{tr}(\mathbf{K}_{MM}^{-1}\Psi_2)]$$
$$+ \beta\sum_{d=1}^{D}\mathrm{tr}(\mathbf{K}_{MM}^{-1}\boldsymbol{\Sigma}_d\mathbf{K}_{MM}^{-1}\Psi_2),$$
$$\langle r(\mathbf{Y},\mathbf{K}_{NM}\mathbf{K}_{MM}^{-1}\mathbf{M})\rangle_{q(\mathbf{X})} = r(\mathbf{Y},\Psi_1\mathbf{K}_{MM}^{-1}\mathbf{M}) + \beta\mathrm{tr}(\mathbf{M}^{\mathsf{T}}\mathbf{K}_{MM}^{-1}(\Psi_2 - \Psi_1^{\mathsf{T}}\Psi_1)\mathbf{K}_{MM}^{-1}\mathbf{M}).$$
$$(11)$$

We also need to compute the K-L divergence $\mathrm{KL}_{\mathbf{U}} := \mathrm{KL}(q(\mathbf{U})\|p(\mathbf{U}))$

$$\mathrm{KL}_{\mathbf{U}} = \int q(\mathbf{U})\log q(\mathbf{U})d\mathbf{U} - \int q(\mathbf{U})\log p(\mathbf{U})d\mathbf{U} = -\mathcal{H}_q(\mathbf{U}) - \langle\log p(\mathbf{U})\rangle_{q(\mathbf{U})}.$$

Denote by $r = \mathrm{vec}^{\mathsf{T}}(\mathbf{U}-\mathbf{M})^{\mathsf{T}}\mathrm{diag}(\{\boldsymbol{\Sigma}_d\})^{-1}\mathrm{vec}^{\mathsf{T}}(\mathbf{U}-\mathbf{M})$. Then $\log q(\mathbf{U}) = -\frac{1}{2}\sum_{d=1}^{D}\log|\boldsymbol{\Sigma}_d| + \frac{MD}{2}\left(\frac{q}{2}-1\right)\log r - \frac{1}{2}r^{\frac{q}{2}}$. From (Proposition A.1. of Li et al., 2023) we know that $r^{\frac{q}{2}} \sim \chi^2(MD)$. Therefore

$$\mathcal{H}_q(\mathbf{U}) = \frac{1}{2}\sum_{d=1}^{D}\log|\boldsymbol{\Sigma}_d| + \frac{MD}{2}\left(\frac{q}{2}-1\right)\frac{2}{q}\mathcal{H}(\chi^2(MD)) + \frac{MD}{2}$$
$$= \frac{1}{2}\sum_{d=1}^{D}\log|\boldsymbol{\Sigma}_d| + \frac{MD}{2}\left(1-\frac{2}{q}\right)\left[\frac{MD}{2} + \log\left(2\Gamma\left(\frac{MD}{2}\right)\right) + \left(1-\frac{MD}{2}\right)\psi\left(\frac{MD}{2}\right)\right] + \frac{MD}{2}.$$

Denote by $\varphi_0(r) := -\frac{D}{2}\log|\mathbf{K}_{MM}| + \frac{MD}{2}\left(\frac{q}{2}-1\right)\log r - \frac{1}{2}r^{\frac{q}{2}}$. Then by Jensen's inequality again

$$\langle\log p(\mathbf{U})\rangle_{q(\mathbf{U})} = \langle\varphi_0(\mathrm{tr}(\mathbf{U}^{\mathsf{T}}\mathbf{K}_{MM}^{-1}\mathbf{U}))\rangle_{q(\mathbf{U})} \geq \varphi_0(\langle\mathrm{tr}(\mathbf{U}^{\mathsf{T}}\mathbf{K}_{MM}^{-1}\mathbf{U})\rangle_{q(\mathbf{U})}),$$

$$\langle\mathrm{tr}(\mathbf{U}^{\mathsf{T}}\mathbf{K}_{MM}^{-1}\mathbf{U})\rangle_{q(\mathbf{U})} = \mathrm{tr}(\mathbf{M}^{\mathsf{T}}\mathbf{K}_{MM}^{-1}\mathbf{M}) + \sum_{d=1}^{D}\mathrm{tr}(\boldsymbol{\Sigma}_d\mathbf{K}_{MM}^{-1}).$$

The elements of $\psi_0$, $\Psi_1$ and $\Psi_2$ can be computed as

$$\psi_0^n = \int k(\mathbf{x}_n, \mathbf{x}_n)\mathrm{q\text{-}ED}(\mathbf{x}_n|\boldsymbol{\mu}_n, \mathbf{S}_n)d\mathbf{x}_n,$$

$$(\Psi_1)_{nm} = \int k(\mathbf{x}_n, \mathbf{z}_m)\mathrm{q\text{-}ED}(\mathbf{x}_n|\boldsymbol{\mu}_n, \mathbf{S}_n)d\mathbf{x}_n,$$

$$(\Psi_2^n)_{mm'} = \int k(\mathbf{x}_n, \mathbf{z}_m)k(\mathbf{z}_{m'}, \mathbf{x}_n)\mathrm{q\text{-}ED}(\mathbf{x}_n|\boldsymbol{\mu}_n, \mathbf{S}_n)d\mathbf{x}_n.$$

With ARD SE kernel (5), we have $\psi_0 = N\alpha^{-1}$. While the integration in $\Psi_1$ and $\Psi_2$ is intractable in general, we can compute them using Monte Carlo approximation. Alternatively, we approximate

$$(\Psi_1)_{nm} \approx \alpha^{-1}\exp\left\{-\frac{1}{2}\langle(\mathbf{x}_n - \mathbf{z}_m)^{\mathsf{T}}\mathrm{diag}(\boldsymbol{\gamma})(\mathbf{x}_n - \mathbf{z}_m)\rangle_{q(\mathbf{x}_n)}\right\}$$

$$= \alpha^{-1}\exp\left\{-\frac{1}{2}[(\boldsymbol{\mu}_n - \mathbf{z}_m)^{\mathsf{T}}\mathrm{diag}(\boldsymbol{\gamma})(\boldsymbol{\mu}_n - \mathbf{z}_m) + \mathrm{tr}(\mathrm{diag}(\boldsymbol{\gamma})\mathbf{S}_n)]\right\},$$

$$(\Psi_2^n)_{mm'} \approx \alpha^{-2}\exp\left\{-\frac{1}{2}\sum_{\tilde{m}=m,m'}(\boldsymbol{\mu}_n - \mathbf{z}_{\tilde{m}})^{\mathsf{T}}\mathrm{diag}(\boldsymbol{\gamma})(\boldsymbol{\mu}_n - \mathbf{z}_{\tilde{m}})) + \mathrm{tr}(\mathrm{diag}(\boldsymbol{\gamma})\mathbf{S}_n)\right\}.$$

If we use the ARD linear form, $k(\mathbf{x}, \mathbf{x}') = \mathbf{x}^{\mathsf{T}}\mathrm{diag}(\boldsymbol{\gamma})\mathbf{x}'$, then we have

$$\psi_0^n = \mathrm{tr}(\mathrm{diag}(\boldsymbol{\gamma})(\boldsymbol{\mu}_n\boldsymbol{\mu}_n^{\mathsf{T}} + \mathbf{S}_n)), \quad (\Psi_1)_{nm} = \boldsymbol{\mu}_n^{\mathsf{T}}\mathrm{diag}(\boldsymbol{\gamma})\mathbf{z}_m,$$

$$(\Psi_2^n)_{mm'} = \mathbf{z}_m^{\mathsf{T}}\mathrm{diag}(\boldsymbol{\gamma})(\boldsymbol{\mu}_n\boldsymbol{\mu}_n^{\mathsf{T}} + \mathbf{S}_n)\mathrm{diag}(\boldsymbol{\gamma})\mathbf{z}_{m'}.$$

A.1.2. LOWER BOUND FOR THE K-L DIVERGENCE ADDED TERMS

Lastly, we need to compute the K-L divergence

$$\mathrm{KL}(q(\mathbf{X})\|p(\mathbf{X})) = \int q(\mathbf{X})\log q(\mathbf{X})d\mathbf{X} - \int q(\mathbf{X})\log p(\mathbf{X})d\mathbf{X} = -\mathcal{H}_q(\mathbf{X}) - \langle\log p(\mathbf{X})\rangle_{q(\mathbf{X})}.$$

Denote by $r = \mathrm{vec}(\mathbf{X} - \boldsymbol{\mu})^{\mathsf{T}}\mathrm{diag}(\{\mathbf{S}_n\})^{-1}\mathrm{vec}(\mathbf{X} - \boldsymbol{\mu})$. Then $\log q(\mathbf{X}) = -\frac{1}{2}\sum_{n=1}^{N}\log|\mathbf{S}_n| + \frac{NQ}{2}\left(\frac{q}{2}-1\right)\log r - \frac{1}{2}r^{\frac{q}{2}}$. From (Proposition A.1. of Li et al., 2023) we know that $r^{\frac{q}{2}} \sim \chi^2(NQ)$. Therefore

$$\mathcal{H}_q(\mathbf{X}) = \frac{1}{2}\sum_{n=1}^{N}\log|\mathbf{S}_n| + \frac{NQ}{2}\left(\frac{q}{2}-1\right)\frac{2}{q}\mathcal{H}(\chi^2(NQ)) + \frac{NQ}{2}$$

$$= \frac{1}{2}\sum_{n=1}^{N}\log|\mathbf{S}_n| + \frac{NQ}{2}\left(1-\frac{2}{q}\right)\left[\frac{NQ}{2} + \log\left(2\Gamma\left(\frac{NQ}{2}\right)\right) + \left(1-\frac{NQ}{2}\right)\psi\left(\frac{NQ}{2}\right)\right] + \frac{NQ}{2}.$$

Denote by $\varphi_0(r) := \frac{NQ}{2}\left(\frac{q}{2}-1\right)\log r - \frac{1}{2}r^{\frac{q}{2}}$. Then similarly by Jensen's inequality

$$\langle\log p(\mathbf{X})\rangle_{q(\mathbf{X})} = \langle\varphi_0(\mathrm{tr}(\mathbf{X}^{\mathsf{T}}\mathbf{X}))\rangle_{q(\mathbf{X})} \geq \varphi_0(\langle\mathrm{tr}(\mathbf{X}^{\mathsf{T}}\mathbf{X})\rangle_{q(\mathbf{X})}),$$

$$\langle\mathrm{tr}(\mathbf{X}^{\mathsf{T}}\mathbf{X})\rangle_{q(\mathbf{X})} = \mathrm{tr}(\boldsymbol{\mu}^{\mathsf{T}}\boldsymbol{\mu}) + \sum_{n=1}^{N}\mathrm{tr}(\mathbf{S}_n).$$

## A.2. Deep Q-EP

We only consider the hierarchy of two QEP-LVMs because the general $L$-layers follows by induction:

$$\begin{aligned} y_{nd} &= f_d^Y(\mathbf{x}_n) + \varepsilon_{nd}^Y, \quad d = 1, \cdots, D, \quad \mathbf{x}_n \in \mathbb{R}^Q, \\ x_{nq} &= f_q^X(\mathbf{z}_n) + \varepsilon_{nq}^X, \quad q = 1, \cdots, Q, \quad \mathbf{z}_n \in \mathbb{R}^{Q_Z}, \end{aligned} \tag{12}$$

where $\boldsymbol{\varepsilon}^Y \sim$ q-ED$(\mathbf{0}, \Gamma^Y)$, $\boldsymbol{\varepsilon}^X \sim$ q-ED$(\mathbf{0}, \Gamma^X)$, $f^Y \sim$ q-$\mathcal{EP}(0, k^Y)$ and $f^X \sim$ q-$\mathcal{EP}(0, k^X)$. Consider the prior $\mathbf{Z} \sim$ q-ED$(\mathbf{0}, \mathbf{I}_{NQ_Z})$. The variational inference for $p(\mathbf{Z}|\mathbf{Y})$ requires maximizing the following ELBO

$$\log p(\mathbf{Y}) \geq \mathcal{L}(\mathcal{Q}) := \int_{\mathbf{Z},\mathbf{F}^X,\mathbf{X},\mathbf{F}^Y} \mathcal{Q}\log\frac{p(\mathbf{Y},\mathbf{F}^Y,\mathbf{X},\mathbf{F}^X,\mathbf{Z})}{\mathcal{Q}}, \tag{13}$$

where the joint probability can be decomposed

$$p(\mathbf{Y},\mathbf{F}^Y,\mathbf{X},\mathbf{F}^X,\mathbf{Z}) = p(\mathbf{Y}|\mathbf{F}^Y)p(\mathbf{F}^Y|\mathbf{X})\cdot p(\mathbf{X}|\mathbf{F}^X)p(\mathbf{F}^X|\mathbf{Z})p(\mathbf{Z})$$

Similarly as in Section 3.1, sparse variational approximation (Titsias and Lawrence, 2010) is adopted to introduce inducing points $\tilde{\mathbf{X}} \in \mathbb{R}^{M\times Q}, \tilde{\mathbf{Z}} \in \mathbb{R}^{M\times Q_Z}$ with associated function values $\mathbf{U}^Y \in \mathbb{R}^{M\times D}, \mathbf{U}^X \in \mathbb{R}^{M\times Q}$ respectively. Hence the augmented probability replaces the joint probability:

$$\begin{aligned} p(\mathbf{Y},\mathbf{F}^Y,\mathbf{X},\mathbf{F}^X,\mathbf{Z},\mathbf{U}^Y,\mathbf{U}^X) =& p(\mathbf{Y}|\mathbf{F}^Y)p(\mathbf{F}^Y|\mathbf{U}^Y,\mathbf{X})p(\mathbf{U}^Y|\tilde{\mathbf{X}})\cdot \\ & p(\mathbf{X}|\mathbf{F}^X)p(\mathbf{F}^X|\mathbf{U}^X,\mathbf{Z})p(\mathbf{U}^X|\tilde{\mathbf{Z}})p(\mathbf{Z}), \end{aligned}$$

where $\mathbf{F}^Y$ and $\mathbf{U}^Y$ are drawn from the same Q-EP; and similarly are $\mathbf{F}^X$ and $\mathbf{U}^X$. Now we specify the approximation distribution as

$$\mathcal{Q} = p(\mathbf{F}^Y|\mathbf{U}^Y,\mathbf{X})q(\mathbf{U}^Y)q(\mathbf{X})\cdot p(\mathbf{F}^X|\mathbf{U}^X,\mathbf{Z})q(\mathbf{U}^X)q(\mathbf{Z}).$$

and choose $q(\mathbf{U}^Y)$ and $q(\mathbf{U}^X)$, and $q(\mathbf{X})$ and $q(\mathbf{Z})$ to be uncorrelated q-ED's:

$$\begin{aligned} q(\mathbf{U}^Y) &\sim \text{q-ED}(\mathbf{M}^Y,\mathrm{diag}(\{\boldsymbol{\Sigma}_d^Y\})), \quad q(\mathbf{U}^X) \sim \text{q-ED}(\mathbf{M}^X,\mathrm{diag}(\{\boldsymbol{\Sigma}_d^X\})), \\ q(\mathbf{X}) &\sim \text{q-ED}(\boldsymbol{\mu}^X,\mathrm{diag}(\{\mathbf{S}_n^X\})), \quad q(\mathbf{Z}) \sim \text{q-ED}(\boldsymbol{\mu}^Z,\mathrm{diag}(\{\mathbf{S}_n^Z\})). \end{aligned}$$

Then the ELBO (13) becomes

$$\begin{aligned} \mathcal{L}(\mathcal{Q}) &:= \int_{\mathbf{Z},\mathbf{U}^X,\mathbf{F}^X,\mathbf{X},\mathbf{U}^Y,\mathbf{F}^Y} \mathcal{Q}\log\frac{p(\mathbf{Y}|\mathbf{F}^Y)p(\mathbf{U}^Y)p(\mathbf{X}|\mathbf{F}^X)p(\mathbf{U}^X)p(\mathbf{Z})}{q(\mathbf{U}^Y)q(\mathbf{X})q(\mathbf{U}^X)q(\mathbf{Z})} \\ &= h(\mathbf{Y},\mathbf{X}) - \mathrm{KL}_{\mathbf{U}^Y} + h(\mathbf{X},\mathbf{Y}) - \mathrm{KL}_{\mathbf{U}^X} + \mathcal{H}_q(\mathbf{X}) - \mathrm{KL}_{\mathbf{Z}}, \end{aligned}$$

where we have

$$h(\mathbf{Y}, \mathbf{X}) = \left\langle \log p(\mathbf{Y}|\mathbf{F}^Y) \right\rangle_{q(\mathbf{F}^Y)q(\mathbf{X})}, \quad h(\mathbf{X}, \mathbf{Z}) = \left\langle \log p(\mathbf{X}|\mathbf{F}^X) \right\rangle_{q(\mathbf{F}^X)q(\mathbf{X})q(\mathbf{Z})}.$$

Note, $h(\mathbf{Y}, \mathbf{X}) \geq h^*(\mathbf{Y}, \mathbf{X})$ is the same as in the bound (7) for Bayesian LVM. However, $h(\mathbf{X}, \mathbf{Z})$ has an extra integration with respect to $q(\mathbf{X})$. Replacing $\mathbf{X}$ with $\mathbf{Z}$ and $\mathbf{Y}$ with $\mathbf{X}$ in (11), we compute

$$\langle r(\mathbf{X}, \Psi_1(\mathbf{K}_{MM}^X)^{-1}\mathbf{U}^X) \rangle_{q(\mathbf{X})} = r(\boldsymbol{\mu}^X, \Psi_1(\mathbf{K}_{MM}^X)^{-1}\mathbf{U}^X) + \mathrm{tr}((\mathbf{I}_D \otimes (\Gamma^X)^{-1}) \, \mathrm{diag}(\{\mathbf{S}_n^X\})).$$

Therefore we have a updated bound for $h(\mathbf{X}, \mathbf{Z}) \geq h^*(\mathbf{X}, \mathbf{Z}) = \varphi(r_{\boldsymbol{\mu}^X}; \Gamma^X, Q)$, where

$$
\begin{aligned}
r_{\boldsymbol{\mu}^X} =\, & r(\boldsymbol{\mu}^X, \Psi_1(\mathbf{K}_{MM}^X)^{-1}\mathbf{M}^X) + \mathrm{tr}((\mathbf{M}^X)^\mathsf{T}(\mathbf{K}_{MM}^X)^{-1}(\Psi_2^X - \Psi_1^\mathsf{T}(\Gamma^X)^{-1}\Psi_1)(\mathbf{K}_{MM}^X)^{-1}\mathbf{M}^X) \\
& + Q[\psi_0 - \mathrm{tr}((\mathbf{K}_{MM}^X)^{-1}\Psi_2^X)] + \sum_{d=1}^{Q} \mathrm{tr}((\mathbf{K}_{MM}^X)^{-1}\boldsymbol{\Sigma}_d^X(\mathbf{K}_{MM}^X)^{-1}\Psi_2^X) \\
& + \mathrm{tr}((\mathbf{I}_Q \otimes (\Gamma^X)^{-1}) \, \mathrm{diag}(\{\mathbf{S}_n^X\})).
\end{aligned}
$$

Finally, we have

$$\mathcal{H}_q(\mathbf{X}) \geq \frac{1}{2}\sum_{n=1}^{N} \log|\mathbf{S}_n^X|, \quad -\mathrm{KL}(q(\mathbf{Z})\|p(\mathbf{Z})) \geq \frac{1}{2}\sum_{n=1}^{N} \log|\mathbf{S}_n^Z| + \varphi_0(\mathrm{tr}((\boldsymbol{\mu}^Z)^\mathsf{T}\boldsymbol{\mu}^Z) + \sum_{n=1}^{N} \mathrm{tr}(\mathbf{S}_n^Z)),$$

where $\varphi_0(r) := \frac{NQ_Z}{2}\left(\frac{q}{2} - 1\right)\log r - \frac{1}{2}r^{\frac{q}{2}}$.

## Appendix B. More Numerical Results

### B.1. Time Series Regression

Table B.1: Regression on simulated time series: mean of absolute error (MAE), standard deviation of predictive distribution (PSD), coefficient of determination ($R^2$), negative logarithm of marginal likelihood (NLL) and running time by various deep models. Each result of the upper part is averaged over 10 experiments with different random seeds; values after $\pm$ are standard errors of these repeated experiments.

| Model | MAE | PSD | $R^2$ | NLL | time |
|---|---|---|---|---|---|
| Deep GP | $0.058 \pm 0.040$ | $0.180 \pm 0.051$ | $0.951 \pm 0.061$ | $-1.432 \pm 0.617$ | $48.409 \pm 0.369$ |
| Deep QEP | $\mathbf{0.049 \pm 0.007}$ | $\mathbf{0.087 \pm 0.004}$ | $\mathbf{0.977 \pm 0.008}$ | $-2.265 \pm 0.158$ | $49.332 \pm 0.475$ |
| DKL-GP | $0.108 \pm 0.135$ | $0.163 \pm 0.039$ | $0.830 \pm 0.382$ | $0.102 \pm 5.055$ | $9.978 \pm 0.172$ |
| DSPP | $0.070 \pm 0.047$ | $0.249 \pm 0.062$ | $0.929 \pm 0.094$ | $-2.192 \pm 0.684$ | $36.495 \pm 0.784$ |

### B.2. Simulated Classification

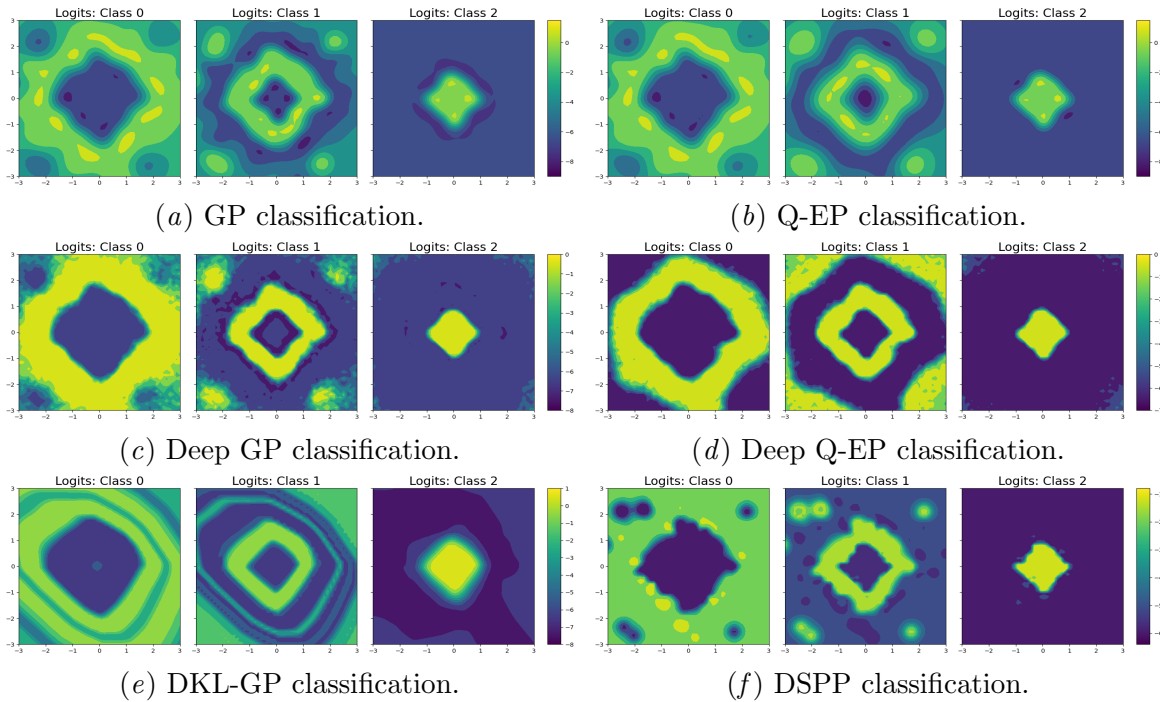

$(a)$ GP classification.

$(b)$ Q-EP classification.

$(c)$ Deep GP classification.

$(d)$ Deep Q-EP classification.

$(e)$ DKL-GP classification.

$(f)$ DSPP classification.

Figure B.1: Comparing Q-EP (B.1($b$)) and deep Q-EP (B.1($d$)) with GP (B.1($a$)), deep GP (B.1($c$)), DKL-GP (B.1($e$)) and DSPP (B.1($f$)) on a classification problem defined on annular rhombus.

Table B.2: Classification on simulated annual rhombus: accuracy (ACC), area under ROC curve (AUC), negative logarithm of marginal likelihood (NLL) and running time by various deep models. Each result of the upper part is averaged over 10 experiments with different random seeds; values after $\pm$ are standard errors of these repeated experiments.

| Model | ACC | AUC | NLL | time |
|---|---|---|---|---|
| GP | $0.810 \pm 0$ | $\mathbf{0.940} \pm 0$ | $17.673 \pm 0$ | $20.622 \pm 0.346$ |
| Deep GP | $0.825 \pm 0.026$ | $0.905 \pm 0.012$ | $534.782 \pm 69.768$ | $124.486 \pm 2.978$ |
| QEP | $0.834 \pm 0$ | $0.935 \pm 0$ | $4.670 \pm 0$ | $20.442 \pm 0.559$ |
| Deep QEP | $\mathbf{0.856} \pm 0.015$ | $0.878 \pm 0.019$ | $96.736 \pm 7.865$ | $124.752 \pm 0.575$ |
| DKL-GP | $0.664 \pm 0.196$ | $0.732 \pm 0.200$ | $17.094 \pm 5.533$ | $23.874 \pm 0.316$ |
| DSPP | $0.744 \pm 0.023$ | $0.829 \pm 0.056$ | $588.543 \pm 302.576$ | $108.076 \pm 1.725$ |

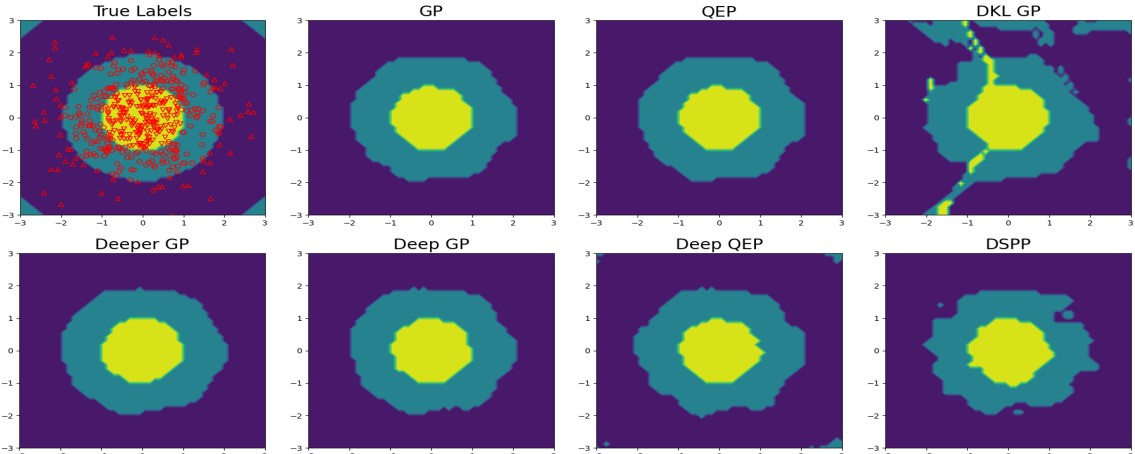

Figure B.2: Comparing shallow (1-layer) and deep (2-layer) Q-EPs with GP, deep GP, deeper GP (3-layer), DKL-GP and DSPP on a classification problem defined on annulus. Circles, upper and lower triangles label three classes in the training data.

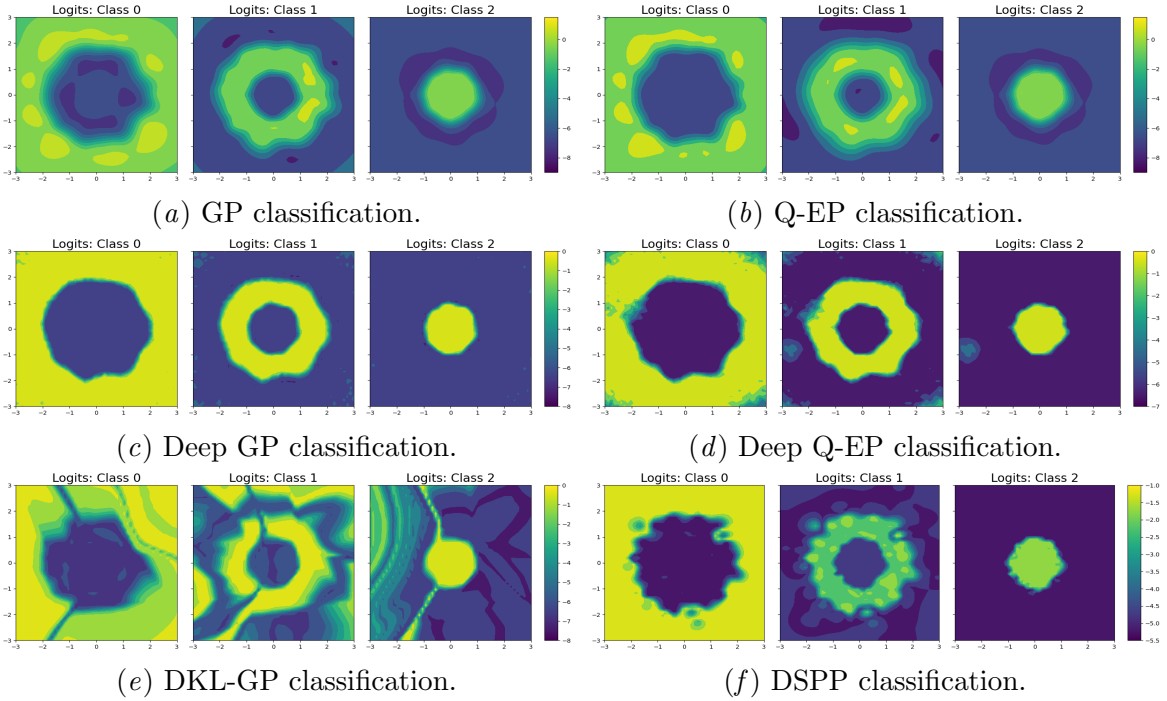

Figure B.3: Comparing Q-EP (B.3($b$)) and deep Q-EP (B.3($d$)) with GP (B.3($a$)), deep GP (B.3($c$)), DKL-GP (B.3($e$)) and DSPP (B.3($f$)) on a classification problem defined on annulus.

Table B.3: Classification on simulated annulus: accuracy (ACC), area under ROC curve (AUC), negative logarithm of marginal likelihood (NLL) and running time by various deep models. Each result of the upper part is averaged over 10 experiments with different random seeds; values after $\pm$ are standard errors of these repeated experiments.

| Model | ACC | AUC | NLL | time |
|---|---|---|---|---|
| GP | $0.951 \pm 0$ | $0.989 \pm 0$ | $18.821 \pm 0$ | $49.425 \pm 1.728$ |
| Deep GP | $\mathbf{0.953} \pm 0.03$ | $0.991 \pm 0.001$ | $467.216 \pm 45.845$ | $199.600 \pm 10.871$ |
| QEP | $0.952 \pm 0$ | $0.985 \pm 0$ | $4.598 \pm 0$ | $49.301 \pm 1.283$ |
| Deep QEP | $0.950 \pm 0.008$ | $\mathbf{0.992} \pm 0.003$ | $123.726 \pm 12.965$ | $197.677 \pm 12.354$ |
| DKL-GP | $0.854 \pm 0.080$ | $0.941 \pm 0.099$ | $19.039 \pm 4.223$ | $34.329 \pm 0.918$ |
| DSPP | $0.922 \pm 0.026$ | $0.970 \pm 0.008$ | $621.152 \pm 297.205$ | $166.974 \pm 2.839$ |

## B.3. Computer Tomography Reconstruction

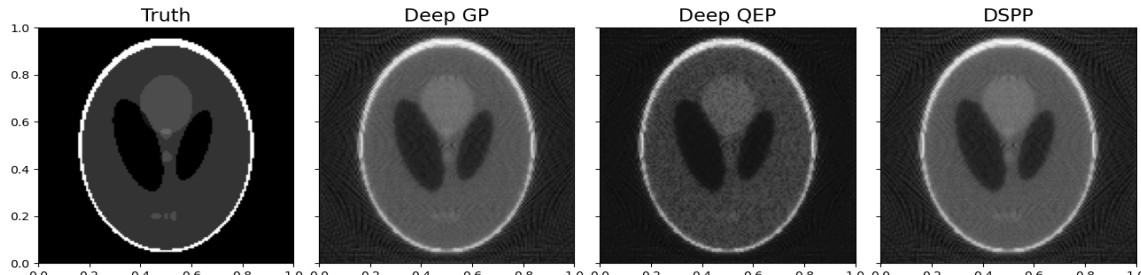

Figure B.4: Shepp-Logan phantom: true image and estimates by deep GP, deep QEP and DSPP.

