# OpenReview forum: "Deep Q-Exponential Processes"
_approximateinference.org/AABI/2025/Proceedings_Track — AABI 2025 Proceedings Track_

### Official Review · Reviewer_uJ6s · 2025-02-21
**Deep Q-Exponential Processes: Advancing Deep Probabilistic Models with Flexible L<sup>q</sup> Regularization for Inhomogeneous Data**

**Rating:** 7
**Confidence:** 4

**Review:**

The paper extends traditional Gaussian processes (GPs) to a more flexible family via a Q-exponential process (Q-EP), which introduces an additional parameter q  to control the regularization. By stacking multiple Q-EP layers, the authors construct a deep variant—deep Q-EP—that not only subsumes deep GPs (when q=2) but also offers better adaptability for modeling inhomogeneous data (e.g., data with sharp edges or abrupt changes). The work covers theoretical derivations, variational inference, sparse approximations, and extensive numerical experiments comparing deep Q-EP with other state-of-the-art deep probabilistic models.

---

### Official Review · Reviewer_h3La · 2025-02-28
**The article presents a generalization of a Q-exponential process to a deep Q-exponential process. The Q-exponential process in the literature is characterized by a variable q which reduces to a Gaussian process for q=2, and is more general for other values of q.**

**Rating:** 6
**Confidence:** 4

**Review:**

The quality of the article is good and the presentation is clear. However, I would not rate it very high on originality as it seems to be a straightforward task to generalize the Q-exponential process to a deep Q-exponential process. The major contribution is the derivation of expressions for evidence lower bound (ELBO) for the shallow Q-exponential and the deep Q-exponential processes.

The significance for deep Q-exponential processes remains the same as for the Q-exponential ones, in that they are both capable of capturing abrupt changes in the datasets with the caveat that deep Q-exponential processes may handle these changes better than their shallow counterparts.

Major points:

1. Section 5.2, what was the reason to sample $x_i$ from a normal and not uniform distribution? Could you repeat the exercise with uniform samples?

2. Page 7, Section 4,
should it be $U^l=[f^l_d(\tilde{X}^{l+1})]_{d=1}^{D_l}$  and not

$U^l=[f^l_d(\tilde{X}^{l})]_{d=1}^{D_l}$. It would be great if you could provide some details for this expansion in the appendix because the definition of $U^l$ is a bit confusing to me.

Minor edits/corrections in no particular order:
1. Page 2, remove point 3. from the contributions to deep probabilistic models as it is not an additional contribution in my opinion but only proof that the first two hold.

2. Possible typo on Pg 5, paragraph 2 of Section 3, 2nd line: should it be ‘where $R \sim...$’ and not ‘where $R^q \sim ...$’?

3. Possible typo on Pg 8, the differential element is missing in the integral term in ELBO.

4. Possible typo on Pg 8, two lines just above Section 5: should there be an asterisk in both $KL_{U^l}$ and $KL_Z$?

5. It maybe better to have all the ELBO loss plots on the same scale (perhaps with a grid) in Figure 2.

6. It would be useful to see a comparison of computational costs across the different methods in Section 5.

---

### Official Review · Reviewer_4Gc3 · 2025-03-01
**Great work**

**Rating:** 7
**Confidence:** 4

**Review:**

Good work

---

### Official Review · Reviewer_ESUU · 2025-03-01
**An interesting contribution with some weaknesses**

**Rating:** 5
**Confidence:** 4

**Review:**

The paper proposes a deep Q-exponential process. This is accomplished by replacing the Gaussian process by the recently introduced Q-exponential process (Q-EP). The paper introduces a sparse variational approximation to facilitate inference. The derivation is more complicated than the derivation for Gaussian processes because several expectations in the evidence lower bound are intractable. The paper sidesteps this issue by applying Jensen’s inequality twice. The paper includes experiments on relatively small-scale datasets, with comparison limited primarily to the Deep GP, DSPP and DKL-GP. The experiments provide some evidence that the proposed method slightly outperforms, particularly when there are step-changes, corners or boundaries.

Strengths

1)	The paper is well-written and presents the method clearly. The structure of the paper is good, exhibiting a logical flow.

2)	The paper introduces a novel derivation of a variational approximation for the presented Q-EP model. Although most aspects of this derivation follow standard developments, there are some additional technical challenges.

3)	The extension of the Q-EP to a deep Q-EP via stacking is a novel proposal.

4)	The presented experiments provide some evidence that the proposed method outperforms the deep-GP in settings where it is expected to.

Weaknesses

1)	There is limited novelty in the sense that the paper essentially proposes the construction of  a deep Q-EP by replacing the Gaussian process by the Q-EP.

2)	Although there are some additional technical challenges in the derivation of the variational approximation, these are addressed by application of Jensen’s inequality. So, yes, there is some technical work, but it is limited in nature.

3)	The experiments involve synthetic and small-scale datasets. There is comparison to a limited set of competing techniques. While there are small performance improvements, the experimental study is not a strong and compelling contribution of the paper.

Summary

Although the paper is well-written and proposes a novel approach, the method is a very simple modification of the deep Gaussian process. The paper does provide a novel variational approximation, but the derivation is mostly straightforward, with the exception of a couple of applications of Jensen’s inequality to replace intractable expectations with lower bounds. The experimental section is relatively weak, revisiting the classic UCI datasets and focusing on synthetic and toy experiments. While such experiments have their place, and should be included in an experimental study, a thorough investigation of the utility of the method would incorporate application to some modern, more challenging benchmarks.

---

### Official Review · Reviewer_actY · 2025-03-03
**An interesting idea, presentation could be streamlined, and experimental discussion could have been more insightful**

**Rating:** 6
**Confidence:** 3

**Review:**

__Quality__

I found that the paper was of overall decent quality, though I thought that the main text could be streamlined to improve readability, and the discussion of the experimental results could be improved.
Some of the presentation could also be improved, for example:
1. Tables 1 & 2 should include error bars (confidence intervals) in the evaluation metrics.
2. I would have liked to have seen the performance of the “shallow GP” model compared against the deep models.

Note: I have not carefully checked the derivations in the paper.

__Clarity__

The clarity of the paper could be somewhat improved by deferring some additional details from the main text (e.g. equation 7) from the main text to the appendix.

Some of the discussion points in the experimental section could be a little clearer, and go a bit further beyond re-iterating results that can be seen in the tables and figures, to discussing where the differences in model performance stem from in the first place.
For example while the q-EP seems to outperform other methods in the synthetic experiments, it seems to be doing markedly worse in several of the tomography experiments (at least for several different depths), but there is no discussion on this point.

A related point on the experiment discussion: the authors assert that “marginal likelihood (NLL) values are not comparable among different models (with distinct probability distributions) and are only listed for reference.”
I would have thought that the marginal likelihood is exactly the metric that is / should be used to compare across models.

__Originality__

The main idea of the paper is to formulate deep q-exponential processes, a model which consists of stacking several q-exponential processes in a way that is similar to how deep GPs are constructed by stacking several GPs.
The overall idea therefore seems analogous to an existing one but seems otherwise original and interesting.

__Significance__

The experimental discussion was somewhat lacking (in terms of explaining differences in model performance) and the conclusions about the performance of the q-EP are therefore (in my view) somewhat limited.
However, the proposed model could be of interest to the wider community.

---

### Meta-Review · Area_Chair_rdkD · 2025-03-16

**Recommendation:** Accept
**Confidence:** 4

**Metareview:**

This paper proposes a deep Q-exponential process (Q-EP) by replacing the Gaussian process (GP) in deep GPs with Q-EP. This makes the model more challenging to work with, and the authors provide derivations to mitigate this issue.

Most reviewers (excluding one-liners) think that this paper is rather straightforward and the experiments are not very extensive. However, in their rebuttal, the authors are confident that they will carefully address these issues. I urge the authors to do so in the paper’s final version.

---

### Decision · Program_Chairs · 2025-03-18

Accept